



# Surface processes in the 7 November 2014 medicane from air-sea coupled high-resolution numerical modelling

Marie-Noëlle Bouin[1,2], Cindy Lebeaupin Brossier[1]

[1]CNRM, Université de Toulouse, Météo-France, CNRS, Toulouse, France
[2]Laboratoire d'Océanographie Physique et Spatiale, Ifremer, University of Brest, CNRS, IRD, Brest, France

*Correspondence to*: Marie-Noëlle Bouin (marie-noelle.bouin@meteo.fr)

**Abstract.** A medicane, or Mediterranean cyclone with characteristics similar to tropical cyclones, is simulated using a kilometre-scale ocean-atmosphere coupled modelling platform. A first baroclinic phase of the cyclone leads to strong convective precipitation, with high potential vorticity anomalies aloft due to an upper-level trough. The deepening and
tropicalization of the cyclone is due first to the crossing of the upper-level jet, then to low-level convergence and uplift of conditionally unstable air masses by cold pools, resulting either from rain evaporation or from advection of continental air masses from North Africa. Backtrajectories show that air-sea heat exchanges warm and moisten the low-level inflow feeding the latent heat release during the mature phase of the medicane. However, the impact of ocean-atmosphere coupling on the cyclone track, intensity and lifecycle is very weak, due to a surface cooling one order of magnitude weaker than for tropical
cyclones, even on the area of strong enthalpy fluxes. Isolating the influence of the surface parameters on the surface fluxes at sea during the different phases of the cyclone confirms the impact of the cold pools on the surface processes. The evaporation is controlled mainly by the sea surface temperature and wind, with a significant additional impact of the humidity and temperature at first level during the development phase. The sensible heat flux is influenced mainly by the temperature at first level throughout the whole medicane lifetime. This study shows that the tropical transition, in this case, is
dependent on processes widespread in the Mediterranean Basin, like advection of continental air, rain evaporation, and dry air intrusion.

## 1 Introduction

Medicanes are small-size Mediterranean cyclones presenting, during their mature phase, characteristics similar to those of tropical cyclones including a perfect vertical alignment between the sea level pressure (SLP) minimum and the minimum of
geopotential at all levels, a cloudless and almost windless column at their centre from the surface to the upper level looking like a cyclone eye, spiral rain bands and a large-scale cold anomaly surrounding a smaller warm anomaly around their centre, vertically extending at least up to the mid troposphere (Reale and Atlas, 2001). However, considering their whole characteristics and lifecycle, they differ from their tropical counterparts by many aspects: their intensity is much weaker,



with maximum wind speed generally reaching those of tropical storms, or Category 1 hurricane on the Saffir−Simpson scale
for the most intense of them (Miglietta et al., 2013); their radius of maximum wind ranges typically 60 to 200 km; due to the
enclosed character of the Mediterranean Sea leading rapidly to landfall, and to the limited ocean heat reservoir, the duration
of their mature phase vary from a few hours to 1 to 2 days; they are able to develop and sustain over sea surface temperature
(SST) typically 15 to 23 °C (Tous and Romero, 2013), much colder than the 26 °C threshold of tropical cyclones (Trenberth,
2005); and a baroclinic phase including vertical wind shear and horizontal temperature gradient is necessary to the early
stage of their development and the establishment of deep convection (e.g. Flaounas et al., 2015).

In the last decade, several studies documented their characteristics and conditions of formation, either from satellite
observations (Tous and Romero, 2013), climatological studies (Flaounas et al., 2015), or case studies based on simulations
(Miglietta et al., 2013; 2017; Miglietta and Rotunno, 2019). They isolated several specificities of the Mediterranean tropical-
like cyclones, either related to large-scale thermodynamical conditions, like the regular presence of an upper-level trough
(also know as a PV streamer) bringing cold air and potential vorticity (PV) from higher-latitude regions, or due to the
geography of the basin, like lee waves forming south of the Alps or north of the North African reliefs, the coasts surrounded
by mountains favouring the uplift of deep convection, and relatively warm sea surface waters able to feed and sustain the
process of latent heat release during their mature phase.

Among Mediterranean cyclones, the classification of Hart (2003) established for tropical cyclones and adapted to the
Mediterranean conditions (Picornell et al., 2014), helps to reliably identify warm core, symmetric events. It appears
nevertheless limited when it comes to the detailed and respective roles of upper-level thermodynamics, low-level
baroclinicity, surface heat exchanges and latent heat release and geographical conditions like orographic lifting. Comparative
studies (e.g. Akhtar et al., 2014) show a large diversity of duration, extension and behaviour and suggest that the medicane
category implies in fact various characteristics.

The role of the large-scale environment like the PV streamer and of the associated upper-level jet has been the subject of
several studies. On a case study of the September 2006 medicane, it was shown for the first time that the crossing of the
upper-level jet by the cyclone to the left exit of the jet resulted in a rapid deepening of the surface low-pressure system by
PV transfer from above (Chaboureau et al., 2012). A comparative study based on composite analysis then enabled to
demonstrate the crucial role of the PV streamer in triggering baroclinic instability and the further role of warm conveyor belt
and dry air intrusion in destabilizing the troposphere close to the cyclone centre (Flaounas et al., 2015). Recently, the
ubiquitous presence of PV streamers and upper-level jets and their key role in the baroclinic development of the medicanes
have been confirmed by a study based on simulations and satellite analyses of several cases of intense medicanes (Miglietta
et al., 2017). All these studies emphasized the importance of the large-scale conditions prior to the development of the
cyclone, but they concluded also that, during their lifecycle, medicanes show a large diversity of characteristics and
behaviours. For instance, the evolution of the upper-level PV anomaly itself may result in the PV streamer wrapping around



the cyclone centre, or in a cut-off low disconnected from its large-scale environment, and a PV tower extending continuously from the low troposphere to the tropopause (Miglietta et al., 2017).

Conversely, the investigation of the contribution of surface processes has motivated few studies so far. Some of them aimed at assessing the relative importance of surface heat extraction versus latent heat release and upper-level PV anomaly on the
cyclone development and lifetime, by using adjoint models, factor separation techniques, or turning off selected processes in sensitivity experiments (Reed et al., 2001; Homar et al., 2003; Moscatello et al., 2008; Carrio et al., 2017). They concluded that, whereas the presence of the upper-level trough during the earlier stage of the cyclone and the latent heat release during its developing and mature phases are necessary to its deepening and maintenance, the role of surface heat fluxes is more elusive. Like in tropical cyclones, the latent heat fluxes always dominate the surface enthalpy processes, with sensible heat
fluxes representing 25 to 30 % of the turbulent fluxes prior to the tropical transition, and 15 to 20 % during the mature phase (Pytharoulis, 2018). Early studies using simulations first concluded that low-level instability controlled by surface heat fluxes may be "an important factor of intensification" (Reed et al., 2001, case of January 1982) and that the latent heat extraction from the sea is a "key factor of feeding of the latent-heat release" (Homar et al., 2003, case study of September 1996). Turning off the surface turbulent fluxes during different phases of the cyclone lifecycle brought contrast to this view,
showing that surface enthalpy is probably equally important as latent heat release during the early baroclinic phase of the cyclone, plays only a marginal role during the deepening phase, and is again crucial in maintaining the cyclone intensity during its mature phase (Moscatello et al., 2008, case study of September 2006).

More recently, studies simulating several cyclones suggested that the impact of the surface fluxes on the cyclone are probably case-dependent (Tous and Romero, 2013; Miglietta and Rotunno, 2019). The latter study especially compared the
medicanes of October 1996 (between the Balearic Islands and Sardinia) and December 2005 (north of Libya) simulated using the same modelling platform. Sensitivity studies performed with and without surface fluxes showed contrasted results, which were attributed to the different competing roles played by the WISHE-type mechanisms (Wind Induced Surface Heat Exchange: Emanuel, 1986; Rotunno and Emanuel, 1987) versus baroclinicity in the two cases. In both cases, instability aloft (a PV streamer) plays a strong role in the initiation of the cyclone, but its evolution during the development and mature
phase is different, with wrapping of the PV streamer around the cyclone centre in the first case, evolution to a cut-off low in the second case. The major difference comes however from the role of surface fluxes, bringing heat and moisture to latent heating through a WISHE-like mechanism in the case of October 1996, and being replaced by warm air seclusion in the case of December 2005. The authors concluded that different categories of intensification mechanisms leading to medicanes co-exist, including the two above-mentioned as well as intensification mechanism through interaction with the left exit of an
upper-level jet, after jet crossing, as in the work of Chaboureau et al. (2012). This suggests that mechanisms of transition towards tropical-like cyclones are diverse, especially concerning the role of the air-sea heat exchanges.



As surface fluxes may strongly depend on the SST, a change of the oceanic surface conditions may, in theory, impact the development of a medicane. Several sensitivity studies investigated the impact of a uniform SST change on the cyclone development and lifecycle, for instance to anticipate the possible effect of the Mediterranean surface waters warming due to climate change. Consistent tendencies were obtained on different case studies (Homar et al., 2003, case of September 1996; Miglietta et al., 2011, case of September 2006; Pytharoulis, 2018, case of November 2014; Noyelle et al., 2019, case of October 1996), showing that, as expected, warmer (respectively colder) SSTs lead to more (resp. less) intense cyclones through a change in the surface enthalpy fluxes. However, changes of SST less than $\pm$ 2 °C result in no significant change in the trajectory, duration or intensity of the cyclone.

The impact of coupling atmospheric and oceanic models has been studied mainly using regional climate models on seasonal to interannual time scales. Comparing coupled and non-coupled simulations using a regional climate model showed an impact of the coupling provided the horizontal resolution of the model is at least 0.08 ° (Akhtar et al., 2014). This resolution proved also necessary to reproduce in a realistic way the characteristic processes of medicanes, including warm cores, and strong winds at low level. Coupled simulations resulted in more intense latent and sensible heat surface fluxes, contrasting with what is usually obtained in tropical cyclones due to the strong cooling effect of the cyclone on the sea surface (Schade and Emanuel, 1999; D'Asaro et al., 2007). This can be due to the use of a 1D ocean model and its limited ability to reproduce the oceanic processes responsible of the cooling. The consideration about the resolution needed to observe an impact of the surface processes was confirmed by the results of Gaertner et al. (2017) comparing several simulations at the seasonal scale, both coupled and uncoupled and from several regional climate modelling platforms. No clear impact of the coupling on the cyclones development was evidenced but the authors attributed this lack of impact to the relatively low horizontal resolution of the coupled experiments, between 18 and 50 km. Finally, a case study comparing higher-resolution (5 km) coupled and uncoupled simulations of the medicane of November 2011 showed no strong impact of the surface coupling, with a weak decrease of the SST of 0.1 to 0.3°C and a difference of 2 hPa on the minimum of SLP and 5 m s$^{-1}$ on the surface wind speed (Ricchi et al., 2017). The impact of ocean-atmosphere coupling in high-resolution (~ 1−2 km), convection-resolving modelshas, to the best of our knowledge, not been assessed yet.

In the present study, we assess the feedback of the ocean surface on the atmosphere of the medicane of November 2014 (also known as Qendresa) over the Strait of Sicily and Ionian Sea using a kilometre-scale ocean−atmosphere coupled model. We investigate the role of the surface processes, especially during the transition phase of the medicane, between its initiation as a Mediterranean storm and its maturity as a purely diabatic cyclone, and we examine the role of the different parameters (including SST) controlling these fluxes throughout the lifecycle of the cyclone.

A brief review of the medicane characteristics and evolution, and the description of the modelling tools and simulation strategy are given in Sect. 2. In Section 3, the results of the reference simulation are validated and used to describe the lifecycle of the event and to highlight the different upper-level and surface processes responsible of its development and





evolution. The impact of the coupling and the role of the surface conditions in controlling the air-sea fluxes are assessed in
125 Sect. 4. These results are discussed in Sect. 5, and some conclusions are given.

## 2 Case study and simulations

The case study is the Qendresa medicane that affected the region of Sicily on 7 November 2014. It has been the subject of
several studies based on simulations, either investigating the role of SST anomalies or the impact of uniform SST change
(Pytharoulis, 2018), the respective role of upper-air instability, surface exchanges and latent heat release (Carrio et al., 2017)
130 or the predictability of the event, depending on the initial conditions and horizontal resolution of the model (Cioni et al.,
2018). All those studies showed that the predictability of this event and especially of its trajectory is rather low, even with
high horizontal (1−2 km) and vertical (50 to 80 levels) resolutions of present operational numerical weather prediction
(NWP) platforms.

### 2.1 The 7 November 2014 medicane

135 On 5 and 6 November 2014, an upper-level PV streamer extended from Northern Europe to North Africa, bringing cold air (-
23 °C) and instability aloft. A general cyclonic circulation developed over the Western Mediterranean basin while Eastern
Mediterranean was dominated by high pressures. At low level on 6 November, the cold and warm fronts associated with the
baroclinic disturbance reinforced due to a northward advection of warmer and moist air, from North Africa. The system
moved towards the Sicily Strait and deepened during the night of 6 to 7 November. On the early hours of 7 November, the
upper-level PV trough and the low-level cyclone progressively aligned, reinforcing the PV transfer from above and the low-
level instability. Strong convection developed, with heavy precipitation in the Sicily area. The low-level system rapidly
deepened in the morning of 7 November, with a sudden drop of 8 hPa in 6 hours, and evolved to the quasi-circular structure
of a tropical cyclone with spiral rain bands and a cloudless eye-like centre. The maximum intensity was reached around
12:00 UTC on 7 November north of Lampedusa. The system drifted eastwards slowly weakening during the afternoon with a
first landfall at Malta around 17:00 then moved northeastwards to reach the Sicilian coasts in the evening of 7 November. It
then continued its decay during the following night on the Ionian Sea close to the Sicily coasts, and lost its circular shape and
tropical cyclone appearance around 12:00 UTC on 8 November.

### 2.2 Simulations

Three numerical simulations of the event were performed using the state-of-the-art atmospheric model Meso-NH (Lac et al.,
2018) and the oceanic model NEMO (Madec and the NEMO Team, 2016).



### 2.2.1 Atmospheric model

The non-hydrostatic French research model Meso-NH version 5.3.0 is used in the present study with a fourth-order centered advection scheme for the momentum components and the piecewise parabolic method advection scheme from Colella and Woodward (1984) for the other variables, associated with a leapfrog time scheme. A C grid in the Arakawa convention

(Mesinger and Arakawa, 1976) is used for both horizontal and vertical discretizations, with a conformal projection system of horizontal coordinates. A fourth-order diffusion scheme is applied to the fluctuations of the wind variables, which are defined as the departures from the large-scale values. The turbulence scheme (Cuxart et al., 2000) is based on a 1.5-order closure coming from the system of second-order equations for the turbulent moments derived from Redelsperger and Sommeria (1986) in a one-dimensional simplified form assuming that the horizontal gradients and turbulent fluxes are much smaller

than their vertical counterparts. The mixing length is parameterized according to Bougeault and Lacarrere (1989) who related it to the distance that a parcel with a given turbulent kinetic energy at level $z$ can travel downwards or upwards before being stopped by buoyancy effects. Near the surface, these mixing lengths are modified according to Redelsperger et al. (2001) to match both the Monin−Obukhov similarity laws and the free-stream model constants. The radiative transfer is computed by solving long-wave and short-wave radiative transfers separately using the ECMWF operational radiation code (Morcrette,

1991). The surface fluxes are computed within the SURFEX module (Surface Externalisée, Masson et al., 2013) using over sea the iterative bulk parametrization ECUME (Belamari et al., 2005; Belamari and Pirani, 2007) linking the surface turbulent fluxes to the meteorological gradients and the SST through the appropriate transfer coefficients. The Meso-NH model shares its physical representation of parameters, including the surface fluxes parametrization, with the French operational model AROME (Seity et al., 2011) used for the Météo-France NWP with a current horizontal resolution of 1.3

km.

In the present study, a first atmosphere-only simulation at the horizontal resolution of 4 km has been performed on a larger domain of 3200 km × 2300 km (D1, see Fig. 1). This simulation started at 18:00 UTC the 6 November and lasted 42 h until 12:00 UTC the 8 November. Its initial and boundary conditions come from the ECMWF−IFS (European Centre for Medium-range Weather Forecasts–Integrated Forecast System) operational analyses every 6 h.

As described in the following, this 4 km resolution simulation then provides initial and boundary conditions for simulations on a smaller domain of 900 km × 1280 km (D2, Fig. 1). All simulations on the inner domain D2 share their horizontal resolution (1.33 km) and vertical grid with 55 stretched terrain-following levels, and a time step of 3 s. Atmospheric and surface parameter fields are issued every 30 minutes.





### 2.2.2 Oceanic model

The ocean model used is NEMO (version 3_6) (Madec and the NEMO Team, 2016) with physical parametrizations as follows. The total variance dissipation scheme is used for tracer advection in order to conserve energy and enstrophy (Barnier et al., 2006). The vertical diffusion follows the standard turbulent kinetic energy formulation of NEMO (Blanke and Delecluse, 1993). In case of unstable conditions, a higher diffusivity coefficient of 10 m$^2$ s$^{-1}$ is applied (Lazar et al., 1999). The sea-surface height is a prognostic variable solved thanks to the filtered free-surface scheme of Roullet and Madec

(2000). A no-slip lateral boundary condition is applied and the bottom friction is parameterized by a quadratic function with a coefficient depending on the 2D mean tidal energy (Lyard et al., 2006; Beuvier et al., 2012). The diffusion is applied along iso-neutral surfaces for the tracers using a Laplacian operator with the horizontal eddy diffusivity value $v_h$ of 30 m$^2$ s$^{-1}$. For the dynamics, a bi-Laplacian operator is used with the horizontal viscosity coefficient $\eta_h$ of $-1.10^9$ m$^4$ s$^{-1}$.

The configuration used here is sub-regional and eddy-resolving, with a 1/36° horizontal resolution over an ORCA grid (from

2.2 to 2.6 km resolution) named SICIL36, that was extracted from the MED36 configuration domain (Arsouze et al., 2013) and shares the same physical parametrizations with its "sister" configuration WMED36 (Lebeaupin Brossier et al., 2014; Rainaud et al., 2017). It uses 50 stretched $z$-levels in the vertical, with level thickness ranging from 1 m near the surface to 400 m at the sea bottom (i.e. around 4000 m depth) and a partial step representation of the bottom topography (Barnier et al., 2006). It has 4 open boundaries corresponding to those of the D2 domain shown in Figure 1, and its time step is set to 300 s.

The initial and open boundary conditions come from the global 1/12° resolution PSY2V4R4 daily analyses from Mercator Océan International (Lellouche et al., 2013).

### 2.2.3 Configuration of simulations

The three-hourly outputs of the large-scale simulation on D1 were used as boundary and initial conditions for 3 different simulations on the smaller domain D2, based on the atmospheric and oceanic configurations described previously. These

three simulations start at 00:00 UTC on 7 November and last 36 h until 12:00 UTC on 8 November. The first atmosphere-only simulation called NOCPL used a fixed SST forcing, while the CPL simulation is the two-way coupled simulation between the Meso-NH and NEMO-SICIL36 model. Indeed, in CPL, the SURFEX-OASIS coupling interface (Voldoire et al., 2017) enables to exchange the SST and two-dimensional surface currents from NEMO to Meso-NH and the two components of the momentum flux, the solar and non-solar heat fluxes and the freshwater flux from Meso-NH to NEMO every 15

minutes. To test the respective impact of the surface currents on the atmosphere with respect to the impact of the SST, another coupled simulation has been performed (NOCUR in the following). It is similar to CPL except that the surface currents are not exported from NEMO to Meso-NH.





In order to ensure that the impact of the coupling in the NOCUR and CPL configurations corresponds to the time evolution of the SST rather than to a change in the initial SST field, the SST field (shown in Fig. 2) used as a surface forcing in NOCPL (and kept constant throughout the simulation) is the field produced by the CPL run, 1 h after the beginning of the simulation (i.e. after the initial adjustment of the oceanic model).

## 2.3 Validation

The trajectories of Qendresa obtained in the three different simulations are compared to the best track based on observations (Cioni et al., 2018) in Figure 2. All the simulated trajectories are shifted northwards with respect to the observations since the beginning of the simulations. The mean distance between the simulated and observed trajectories is close to 85 km with no significant difference between the simulations. Cioni et al. (2018) showed that using horizontal resolutions finer than 2.5 km is mandatory to accurately represent the fine-scale structure of this cyclone and its time evolution. Sensitivity studies showed an increased convergence of simulated track towards the observations with higher resolution, the best agreement being obtained with a nested configuration and an inner domain at 300 m resolution. In the present study, several sensitivity tests based on these results were performed on the smaller-domain simulation to improve the simulated trajectory: i) the starting time of the simulation was changed between 12:00 UTC on 6 November and 00:00 UTC on 7 November with increment of 3 h; ii) the number of vertical levels in Meso-NH was increased to 100, with a stretching ensuring a better sampling in the atmospheric boundary layer; iii) the atmospheric simulation was performed without nesting, initial and boundary conditions from ECMWF−IFS, and horizontal resolution of 2 km. None of these tests resulted in a significantly improved trajectory, the northward shifting of the cyclone occurring in every case in the early hours of the 7 November.

The simulated cyclone nevertheless shows a lifecycle and intensity close to observations, even if a direct (i.e. co-localized) comparison is not possible due to the northward shift of its trajectory. A strong deepening of almost 15 hPa is obtained in the first 12 h of the CPL simulation (Fig. 3b) with a minimum value at 12:30 UTC on the 7 November close to the minimum observed at Linosa station. This station has been chosen as the closest point to the best track from observations at the time of the observed maximum intensity of the storm. The surface wind speeds show peak values at the same time (Fig. 3a), and a time evolution in good agreement with METAR observations at the stations of Lampedusa, Pantelleria or Malta (not shown). Wind speed averaged over a 50 km radius around the cyclone centre presents a time evolution close to the control simulation of Cioni et al. (2018).

Phase space diagrams are commonly used to describe in a synthetic way the symmetric characteristics of the cyclone, as well as the thermal characteristics and extent of its core. The present version showing the evolution of Qendresa from 01:00 UTC on 7 November to 12:00 UTC on 8 November (Fig. 4) is derived from the original work of Hart (2003) using the adaptation of Picornell et al. (2014) for smaller-scale cyclones. The radius used for computing the low troposphere thickness asymmetry $B$, the low-troposphere and upper-troposphere thermal winds (-$V_{TL}$ and -$V_{TU}$ respectively) has been fitted to the radius of





maximum wind and is close to 100 km, and the low troposphere and upper troposphere are defined here as the 925−700 hPa
and 700−400 hPa levels respectively. The simulated cyclone presents a symmetric warm core throughout its whole lifetime,
and this warm core moves from shallow to deep between 08:00 and 09:00 on 7 November, remaining deep till the end of the
simulation. This confirms that the simulated cyclone presents the behaviour of a medicane and is very similar, in its lifecycle
and intensity, to the observed one. Despite the shift in its trajectory, the simulation results may be used to investigate the
processes responsible for its deepening and sustaining, and the surface mechanisms in link with the ocean feedback. In the
following, except if otherwise specified, the results of the NOCPL simulation are used for investigating the medicane
behaviour and to focus on what has been presented as the area of interest (AI) in Figure 1.

**3 Medicane lifecycle and coupling impact**

This part presents first the successive phases of the event based on an analysis of upper-level and mid-troposphere processes.
Then, these three phases are described in more detail, with a focus on the surface and low-level mechanisms identified from
the simulation outputs. Finally, the impact of taking into account the short-time evolution of the SST on the atmospheric
surface processes, through ocean−atmosphere coupling, is assessed.

**3.1 Chronology of the simulated event**

On the large-scale simulation at 22:00 UTC on 06 November, the upper-level trough extends over Eastern Algeria, Tunisia
and Western Libya, whereas the SLP minimum is located between Pantelleria and Sicily, under the 300 hPa jet (not shown).
The upper-level PV anomaly and the associated jet shift northeastwards towards Sicily, and at 02:00 UTC, the SLP minimum
is now located under the left exit of the jet (Fig. 5a). This corresponds to the start of the deepening of the cyclone, via
processes similar to those described in Chaboureau et al., (2012), i.e. barotropic instability and upper-level strong horizontal
shear leading to a transfer of vorticity from aloft to mid to lower troposphere. This first deepening phase occurs in a strongly
baroclinic environment ("baroclinic phase" hereafter) with a cold front still present east of the domain at low level (Fig. 6a).
At 08:30 UTC, the jet has further moved over the Ionian Sea and Sicily and the SLP minimum is now very close to the
northwesterm tip of the 300 hPa PV anomaly (Fig. 5b). This marks the beginning of the second, more intense deepening of
the medicane leading to its peak in intensity at 12:00 UTC on 7 November (Fig. 3). It corresponds to the alignment between
low- and upper-level maxima of PV anomalies, and the transition towards the mature, diabatic phase of the medicane. This
phase is called "transition phase" hereafter. At 12:00 UTC, the medicane, which has gained its maximum intensity, presents
the circular shape typical of tropical cyclones with spiral rainbands and no trace of baroclinicity. The upper-level PV
anomaly is now wrapped around the SLP and this vertically lined-up cyclone centre detaches from the upper-level jet, which
is now located over Southern Italy (Fig. 5c). The cyclone then moves northeastwards towards the Ionian Sea and





continuously decreases until 08:00 UTC on 8 November. This last phase is called "diabatic phase" hereafter. An analysis of the distribution and time evolution of the precipitation shows that the heaviest rainfalls occur during the baroclinic and

transition phases, with values above 50 mm h$^{-1}$ locally east of Sicily and at sea between Pantelleria and Malta. During the diabatic phase, rain is more scattered or organized in spiral rainbands and reaches barely 40 to 50 mm h$^{-1}$ locally.

### 3.1.1 Baroclinic phase

At low level, this phase corresponds to low-pressure system resulting of the evolution of the instability generated by the lee wave of the North African relief, with strong baroclinic structures. An analysis of the potential temperature $\theta$ and near-

surface wind field at 04:00 on 7 November (Fig. 6a) shows that baroclinic processes are responsible for the heavy precipitation in the first hours of the event (instantaneous rain rate > 5 mm h$^{-1}$ in green Fig. 6a). A warm sector is present at the east of the domain, with a cold front extending south-east from the south of Italy and very strong low-level convergence between a southeasterly flow in the warm sector and a south to southwesterly flow in the cold sector (white contours when > 1 10$^{-3}$ m s$^{-2}$). This convergence is at the origin of deep convection and results in strong precipitation at sea, as shown by the

east-west cross section close to the cyclonic centre (Fig. 6b). Two bands of colder low-level air masses are present south and east of Sicily, in the east of the domain, and result of the heavy precipitation occurring during the night of the 6 to 7 November. These cold pools have been generated by evaporative cooling typically between 1000 and 1500 m above sea level under strong precipitation. The cold and humid air spreads to the surface following density currents and is advected northeastwards by the low-level flow. The resulting cold pools generate strong horizontal convergence and trigger uplift up

to 2000 m and deep convection of the relatively warm air masses of the southerly flow (Fig. 6b). Along the eastern and southern coasts of Sicily, orographic uplift of the southerly low-level flow also result in heavy precipitation. Cold air masses are also found at low level in the south of the domain, that were formed at night by radiative processes over land then advected over sea from North Africa.

### 3.1.2 Convective transition

At 08:30 UTC on 7 November, no clear baroclinic structure is present at low level (Fig. 7a) but the convective activity responsible for strong precipitation is still present east of the domain as the remaining of the strong convergence described previously, and south of Sicily, close to the cyclonic centre. The surface virtual potential temperature $\theta_v$ superimposed to the equivalent potential temperature $\theta_e$ on the maps (Fig. 7a) and on the north-south (N-S) and east-west (E-W) cross sections close to the SLP minimum (Fig. 7b and 7c) is used here as a marker of cold pools (with an upper limit of 19°C for $\theta_v$ –

Ducrocq et al., 2008; Bresson et al., 2012). Some of these cold pools are the result of evaporating processes under convective precipitation, as above, while those located at sea along the North African coast originates from dry and cold air advected



from inland. The strong horizontal convergence at low level, leading to uplift and deep convection on air masses with high $\theta_e$, is located on the upwind edge of the cold pools (Fig. 7a, 7c). During this transition phase, the cold pools located in the southerly flow move northwards, towards the centre of the cyclone and trigger uplift and deep convection up to 3000 m of

the northwesterly low-level flow with high $\theta_e$ (Fig. 7b). This propagates the surface warm anomaly close to the cyclone centre (which is now located under the 300 hPa PV anomaly) up to 3000 m and develops a corresponding low- to mid-troposphere PV anomaly. At the same time, a dry air intrusion from the upper levels brings air masses with low $\theta_e$ and relative humidity below 20 % to 3000 m, resulting in a upper-to-mid-troposphere PV anomaly. The low-level vortex increases its spin, enhancing low-level convergence and further developing the warm core of the cyclone through latent heat

release. This mechanism is able to sustain itself while the low-level flow of the low-pressure system has a high CAPE (convective available potential energy), obtained by extracting heat and moisture through strong heat fluxes. This aspect will be further discussed in section 4.

### 3.1.3 Diabatic phase

At 13:00 on 7 November, the PV anomalies at 700 hPa and 300 hPa are now aligned (Fig. 8a). A zonal cross section on the

SLP minimum shows that a low-level PV anomaly with values above 5 PVU has formed around the centre of cyclone, extending from the surface to join the 300 hPa anomaly (Fig. 8b). The warm core of the systems extends up to 850 hPa, and is limited upward by the colder air (low $\theta_e$) brought from aloft. The tangential velocity field shows low-level convergence (up to 800 hPa) towards the cyclone centre, deep convection close to the centre, but no or very weak divergence at mid to upper troposphere. The cyclonic circulation has reinforced with horizontal wind speed above 8 m s$^{-1}$ at all heights out of a

radius of 10 km around the cyclone centre. In terms of processes, this phase corresponds to diabatism identified as the main mechanism generating and sustaining tropical cyclones in their mature phase. The low-level convergence of warm and humid air towards the cyclone centre results in convection, maintains the existing vorticity and the cyclonic circulation through latent heat release of the uplifted air masses. In the afternoon of the 7 November, the cyclone first moves towards colder SST in the east of the Sicily Strait (Fig. 2), then made a landfall at the Sicilian south-east corner around 17:00 UTC. It

then reaches the Ionian Sea around 20:00 UTC, with even colder SSTs, before slowly decaying and losing its tropical-like characteristics. To ensure that the warm core of the cyclone is due to latent heating fed by warm and moist air extraction from the sea-surface rather than to warm air seclusion in a baroclinic environment as for the December 2005 medicane (Mazza et al., 2017; Fita and Flaounas, 2018), backtrajectories were used, starting at 23:00 on the 7 November, south of the cyclone centre. The trajectories of three air parcels originating from very different places and arriving at the same place, at

three vertical levels surrounding the level closest to 1500 m, are shown in Fig. 9. Their equivalent potential temperature ranges from 31 to 38 °C at their first appearance in the domain and is close to 45 °C on average when they reach their final point. On their trajectories, $\theta_e$ increases almost continuously, with a strong jump during their transit at low level (below 500





m) above the sea, on the area with enthalpy fluxes above 600 w m$^{-2}$ (EF600 hereafter, white contour in Fig. 9 – the enthalpy flux is defined as the sum of the latent heat flux *LE* and sensible heat flux *H*). This demonstrates the strong role of the sea

surface in increasing the moisture and heat of the low-level flow before its approach of the cyclone centre, and confirms that the warm core of the medicane is actually due to diabatic processes.

In the following, the possible impact of the ocean−atmosphere coupling on the cyclone intensity is examined by comparing the results of the CPL, NOCUR, and NOCPL simulations. The time period for this comparison is the 7 November only, as the medicane has lost a large part of its intensity in the evening of the 7 November.

**3.2 Sensitivity to oceanic coupling**

From Figure 3, it can be seen that the influence of the SST time evolution on the medicane is small. Taking into account the effect of the SST cooling only (NOCUR) results in a slightly slower and less intense deepening by 1.5 hPa and almost no change of the maximum wind (Fig. 3b). Including the effect of the surface currents on the atmospheric boundary layer results is a slightly more intense cyclone (1.5 hPa difference) at its maximum and 8 m s$^{-1}$ stronger maximum wind. Figure 2 shows

also that no significant difference on the trajectory is obtained between the NOCPL, NOCUR and CPL simulations, except maybe when the cyclone centre loops east of Sicily at the end of the day.

In the following, we examine whether this very small sensitivity is related to the evolution of the SST or to the surface processes involved.

**3.2.1 SST evolution**

The median values of the difference of the SST between the CPL and NOCPL simulations over the whole domain, and the values of the 5 %, 25 %, 75 % and 95 % quantiles are shown Figure 10. This median surface cooling is very weak and reaches barely 0.1 °C at the end of the baroclinic phase, and still 0.1 °C at the beginning of the diabatic phase. Its further evolution, during the diabatic phase, is still very weak with values of 0.2 °C at 23:00 UTC, on 07 November. The maximum cooling is 0.6 °C. To focus on the effects of this surface cooling on the surface processes feeding the cyclone, we used a

conditional sampling technique to isolate the areas with enthalpy flux above 600 W m$^{-2}$ that corresponds to the mean value of the 80 % quantile of the enthalpy flux on the day of the 7 November. On this area (EF600), the SST difference and its time evolution are slightly larger with a median difference of -0.2 °C at the beginning of the diabatic phase and close to -0.4 °C at the end of 7 November. The SST difference obtained in NOCUR on EF600 is slightly larger than in CPL but the difference is not significant. The SST cooling on the area of highest fluxes that are responsible for supplying the medicane in heat and

moisture is therefore under 0.4 °C in median value, and much weaker than typical cooling values observed under tropical cyclones, that commonly reach 3 to 4 °C (e.g. Black and Dickey, 2008). In addition, the spatial extent of the cooling does not correspond to a clear wake as in tropical cyclones (not shown). So, the conclusion of this part is that surface cooling under





this medicane is one order of magnitude smaller than what is obtained under tropical cyclone. Quantifying the surface cooling under other medicanes could lead to contrasting results. For instance, in an ocean-atmosphere-waves coupled
simulation of a strong storm in the Gulf of Lion, surface cooling of 2 °C was obtained (Renault et al., 2012).

### 3.2.2 Impact on turbulent surface exchanges

A comparison of the time evolution of the enthalpy flux, sensible and latent heat fluxes of the NOCPL and CPL simulations shows that the differences are very weak even on the EF600 area (Fig. 11a). During the diabatic phase where it is maximum, the mean difference of the enthalpy flux is 24 W m$^{-2}$, with a standard deviation of 12 W m$^{-2}$. Compared to the values of the
turbulent fluxes on this area, between 500 and 800 W m$^{-2}$ for $LE$ and 100 and 250 W m$^{-2}$ for $H$, this value is weak. Expressed in percent of the fluxes values, the relative difference is close to 3 % at the beginning of the diabatic phase and reaches 5 % at 21:00 UTC on 7 November, when the medicane is already weakening with maximum wind speed below 25 m s$^{-1}$. The relative difference of the sensible heat flux varies between 4 and 10 % due to the lower values of $H$, and the value of the difference is close to 7 W m$^{-2}$ with a standard deviation of 4 W m$^{-2}$. So, coupling appears to have a very weak impact on the
turbulent heat fluxes even in the EF600 area. Again, the effect of the surface currents (CPL versus NOCUR in Fig. 11b) is not significant.

In the following, we investigate the time evolution of the surface turbulent fluxes, and the processes controlling them.

### 4 Role of surface fluxes and mechanisms

This section investigates the role of the surface parameters in controlling the surface heat fluxes during the different phases of the medicane. The objective is to assess the relative role of the SST, the surface wind, and the heat and moisture in the surface layer in the surface heat transfer and its time evolution.

### 4.1 Parameterization of surface fluxes

In numerical atmospheric models, the turbulent heat fluxes are classically computed as a function of surface parameters
using bulk formulae:

$$H = \rho\, c_p\, C_h\, \Delta U\, \Delta \theta \qquad\qquad (1)$$

$$LE = \rho\, L_v\, C_e\, \Delta U\, \Delta q \qquad\qquad (2)$$

with the air density, $c_p$ the air thermal capacity and $L_v$ the vaporization heat constant, $\Delta U$, $\Delta \theta$ and $\Delta q$ the gradient of the wind speed, the difference between the SST and the potential temperature at first level, and the difference between the specific





humidity of the SST at saturation and the specific humidity at first level, respectively. The transfer coefficients $C_h$ and $C_e$ are

defined as

$$C_h^{1/2} = \frac{C_{hn}^{1/2}}{1 - \dfrac{C_{hn}^{1/2}}{\kappa} \psi_T(z/L)} \tag{3}$$

and

$$C_e^{1/2} = \frac{C_{en}^{1/2}}{1 - \dfrac{C_{en}^{1/2}}{\kappa} \psi_q(z/L)} \tag{4}$$

with $\kappa$ the von Karman's constant, $\psi_T$ and $\psi_q$ empirical functions describing the stability dependence, $C_{hn}$ and $C_{en}$ the neutral

transfer coefficient for heat and moisture and $L$ the Obukhov length (which depends, in turn, on the virtual potential

temperature at the surface and on the friction velocity $u_*$). In the ECUME parameterization used in this study, the neutral

transfer coefficients $C_{hn}$ and $C_{en}$ are defined as polynomial functions of the 10 m neutral wind speed.

With such parameterizations, the surface air-sea heat fluxes are dependent on the wind speed at the first level of the model,

on the gradients of potential temperature and specific humidity between the sea surface and the first level, and on the transfer

coefficients $C_h$ and $C_e$. The gradient of potential temperature $\Delta\theta$ depends on the potential temperature at first level $\theta$ and on

the SST. The gradient of humidity $\Delta q$ depends on the specific humidity at the first level $q$ and on the SST used to compute

the specific humidity at saturation at the sea surface. The transfer coefficients depends on the wind speed at 10 m and on the

Obukhov length through the stability functions. The Obukhov length is expressed as in Liu et al. (1979):

$$L = -\frac{T_v^2 u_*^2}{\kappa g T_{v*}} \tag{5}$$

with $T_v$ the virtual temperature at the first level, depending on the temperature and specific humidity, and $T_{v*}$ the scale

parameter for virtual temperature depending on the temperature and humidity at the first level. As a consequence, the transfer

coefficients depend as the fluxes on the wind speed, on the temperature and specific humidity at the first level, and on the

SST. In the following, we do not distinguish between the temperature and potential temperature at first level.

The time evolution of the median values, and 5 %, 25 %, 75 % and 95 % quantiles of the latent and sensible heat fluxes is

shown in Figure 12a for the 7 November, on the EF600 area, and the time evolution of the median values and quantiles of

the SST in Figure 12b. The latent heat flux is always much higher than the sensible heat flux, as this is generally the case at

sea when the SST is above 15 °C (e.g. Reale and Atlas, 2001). The sensible heat flux represents here 22 % of the total

turbulent flux during the development of the medicane, 13 to 17 % during the diabatic phase. Both fluxes show asymmetric





distributions with upper tail (95 %) more distant from the median than the lower tail (5 %). This is partly due to the conditional sampling ($LE + H > 600$ W m$^{-2}$) used here. The median value of the sensible heat flux is maximum at the end of the baroclinic phase (180 W m$^{-2}$ at 08:00 UTC), while the maximum value of the 95 % quantile is reached since the beginning of the baroclinic phase at 04:00 UTC, with 332 W m$^{-2}$. From the beginning of the transition phase until 18:00 on 7 November, both the median and 95 % quantile values of the sensible heat flux are continuously decreasing. Conversely, the

maximum of the median value of the latent heat flux (635 W m$^{-2}$) is reached at 09:00 UTC during the transition phase and stays approximately constant until 15:00 UTC, during the diabatic phase. The maximum of the 95 % quantile (845 W m$^{-2}$) is reached at the beginning of the transition phase. The decrease of the latent heat flux starts later than the for the sensible heat flux (around 15:00, as the system is already weakened) and is slower until the end of the 7 November. The median values of $LE$ in this EF600 sampling are constant or slightly increasing until the evening (20:00 UTC), whereas the minimum values (5

% quantile) increase continuously until the end of the day. Again, this is probably partly due to the sampling used here.

The time evolution of the median values and quantiles of the SST shows, conversely, asymmetric distributions with lower tails much longer than upper tails (Fig. 12b). The maximum values of SST (95 % quantile) are almost constant with time and close to 24 °C, while the lower and median values vary due to the conditional sampling EF600 and the motion of the cyclone away from the warm SST area. The time evolution of the distributions of $LE$ and the SST are opposite to each other, the

minimum of SST is observed at the beginning of the transition phase.

To investigate the mutual dependencies and co-variabilities of the fluxes and parameters listed above, we used the rank correlation of Spearman, which corresponds to the Pearson or linear correlation between the rank of the two variables in their respective sampling (Myers et al., 2010). This metrics enables relating monotonically rather than linearly the variables of interest and is more appropriate in the case of non-linear relationships as this is the case for the fluxes that may be related to

the variables additionally through the transfer coefficients.

The following analysis is conducted for the latent and sensible heat fluxes separately, as the controlling parameters and mechanisms are supposedly different. The co-variabilities are analysed first in the whole domain, to determine what contributes most to the strong fluxes globally, then in the EF600 area to isolate processes explicitly responsible for the fluxes contributing the most to the growth and maturity of the medicane. The corresponding values are given in Tables 1 to 3 for the

EF600 area, and for 3 time periods considered representative of the baroclinic, transition and diabatic phases respectively, i.e. 04:00, 10:00 and 17:00 UTC on 7 November.

## 4.2 Latent heat flux

The time evolution of the Spearman's rank correlations between the latent heat flux, $U_{10}$, $\theta$, the SST and $q$ is given Figure 13 and Tables 1 to 3. On the whole domain, the parameters controlling $LE$ (and evaporation) are the SST and the wind speed

throughout the whole duration of the simulated event, with very strong positive correlations.





Within the EF600 domain, the respective influences are contrasted and show a larger variability. All the correlations are globally positive but moderate (see Fig. 13b by contrast with 13a). That means that for the specific humidity or temperature, the control is from the latent heat on the parameters (strong evaporation results in more heat / moisture at low level). The role of the SST is weaker than on the whole domain but still positive, except at the very end of the simulation, while the role of the wind speed is progressively decreasing.

During the baroclinic phase, on the whole domain, the controlling parameters are the SST and the wind (positively correlated), the specific humidity (negatively) and the potential temperature (negatively). During this phase, $\theta$ and $q$ are also strongly positively correlated ($r_s = 0.55$ over the whole domain), due to the advection of cold and dry air by the southerly low-level flow from the Tunisian and Libyan continental surface (Fig. 14b, c and f). This air mass progressively charges itself in heat and moisture on the area of strongest enthalpy fluxes north of the Libyan coasts (Fig. 14a). At that time, the EF600 area of strong fluxes and cold/dry air corresponds also to the area of warm SST (Fig. 14e). Within this area, the main influence on $LE$ is from the wind, then from the SST (Fig. 13b, Table 1). There is no effect of the potential temperature or specific humidity (weak or negative correlations, Fig. 13b, Table 1).

The same is true during the transition phase: over the whole domain, the dominant role is played by the SST with an effect equivalent to the wind speed, and a decreasing influence of the temperature and humidity. The area EF600 extends further north, closer to the cyclone centre, away from the area of cold and dry low-level air, which also tends to warm and moisten under the combined impact of the diurnal warming of the continental surfaces (not shown) and of the strong enthalpy fluxes offshore (Fig. 15a, c and f).

During the diabatic phase, the influence of the humidity on the whole domain is much weaker (Fig. 13a). The area of strong enthalpy fluxes is still located on warm SSTs on the south of the domain (Fig. 16a, e), which is also the place of the strongest winds on the right-hand side of the cyclone (Fig. 16b). Within this EF600 area, there is almost no influence of the temperature or humidity on $LE$. The influence of the wind speed is decreasing, the role of the SST is strong until 21:00 UTC when the cyclone reaches the Ionian Sea where the SST is much colder, and the effect of the wind speed becomes dominant at the very end (Fig. 13b).

In summary, at the scale of the domain, the latent heat flux (evaporation) is controlled by the SST and wind throughout the day of the 7 November: both strong winds (in the cold sector during the baroclinic phase, then close to the cyclone centre and in its right side) and warm SSTs (in the south of the domain) are thus necessary to have strong latent heat fluxes. Within the area where the turbulent fluxes are high (and where winds are strong and SSTs high), the control is mainly from the wind (baroclinic and transition phases) then from the SST (diabatic phase).



### 4.3 Sensible heat flux

The time evolution of the Spearman's rank correlations between the sensible heat flux, $U_{10}$, $\theta$ and the SST is given Figure 17 and Tables 1 to 3. On the whole domain, the parameters controlling the sensible heat transfer are mainly the potential temperature in the surface layer during the baroclinic and transition phases, then the wind (with some influence of the temperature and SST) during the diabatic phase. On the EF600 area, the SST influence is weak at all times, the major control is from the potential temperature, partly indirectly through the stratification and transfer coefficient (not shown). The wind plays a role mainly at the beginning of the diabatic phase and to some extent during the baroclinic and transition phases.

The latent heat flux is always higher than the sensible heat flux (Fig. 12a), resulting in the "strong flux area" EF600 being determined by $LE$ values rather than $H$ values, and more homogeneous values of $LE$ than $H$ over this area. However, the sensible heat flux can reach strong values locally with respect to the latent heat flux, especially during the baroclinic and transition phases. As a consequence, values of the sensible heat flux still show strong contrast on the EF600 area (Fig. 14d, 15d) in these two phases. The enhanced control by the potential temperature during these phases is partly due to the continental air masses advected from North Africa, and partly to the presence of the cold pools under the areas of deep convection and strong wind. During the baroclinic phase (Fig. 14b, c and d), the strong sensible heat fluxes are located offshore of the Tunisian and Libya coasts downwind of the strong low-level flow bringing cold air from the continent. At 10:00 UTC by contrast (Fig. 15c and 15d), sensible heat fluxes above 300 W m$^{-2}$ are collocated with the cold pools that form under convective precipitation close to the cyclone centre and under gust fronts at the south of the domain. During the diabatic phase (Fig. 16), there are less patches of strong sensible heat flux corresponding to cold pools and low $\theta$, but a medium-scale northwest−southeast gradient of $H$ over the EF600 area, related to a NS gradient of wind speed (Fig. 16b) and to a northwest−southeast gradient of potential temperature (Fig. 16c) with warmer temperatures downwind, as the air mass progressively warms inside the eastward continental flow due to strong surface heat fluxes (mainly the latent heat flux). On the north of the EF600 area (where the wind speed is also the highest), the potential temperature is colder and the sensible heat fluxes are maximum.

In summary, the sensible heat flux is always mainly controlled by the potential temperature in the surface layer. Colder air masses result in enhanced sensible heat flux, rather than strong wind or warmer SST. During the two first phases, this cold air is either advected from North Africa or created by evaporation under convective precipitation (cold pools). During the diabatic phase, strong latent heat transfer over warm SSTs warms the near-surface atmospheric layer and results finally in lower sensible heat transfer.



## 5 Discussion and conclusion

The comparison of the simulations with and without coupling with an oceanic model shows no significant impact of the
evolution of the SST on the track, intensity or lifecycle of the medicane. The weak SST cooling, notably during the first 24 h
of the simulation, is likely responsible for that. On the strong flux area, where the enthalpy flux feeds the cyclone in heat and
moisture maintaining the convection and the latent heat release mechanism, the median value of the SST cooling is 0.2°C at
the beginning of the diabatic phase, and reaches barely 0.4 °C at the end of the 7 November. The median difference on $H$ is
-7 W m$^{-2}$ at the beginning of the diabatic phase, -12 W m$^{-2}$ at 23:00 UTC on the 7 November (representing less than 10 %
difference), and -19 W m$^{-2}$, -37 W m$^{-2}$ on $LE$ at the same two time periods (representing less than 5 % difference).
The co-variabilities of surface fluxes and parameters show nevertheless that, in this specific case, the SSTs exert a strong
control on the latent heat flux that dominates the surface heat transfer, throughout the whole duration of the event. During its
development phase, there is also a strong influence of peculiarities of the Central Mediterranean: the transition between deep
convection and heavy precipitation associated to baroclinic (frontal) processes and the tropical-like cyclone takes place
downwind of the low-level flow of dry and cold air originated from North Africa. These air masses with low $\theta_v$ encounter
moist and warm air resulting of the strong sea-surface evaporation and enhance the deep convection at sea, together with the
cold pools formed by rain evaporation and downdrafts. These cold pools of various origins displace the deep convection at
sea. Uplift of warm air masses increases the low-level PV, and reinforces the vortex, which is moved northeastwards closer
to the PV anomaly aloft.
It has recently been suggested that medicanes could be sorted into two (possibly three) different categories according to the
intensity and role of air-sea heat exchanges and to the related surface mechanisms (Miglietta et al., 2019). The main
difference between these two categories is the processes leading to the warm core of the cyclone. The first category
corresponds to WISHE-like mechanisms, with extraction of heat and moisture from the sea surface and latent heat release as
low-level processes responsible for the medicane deepening and warm-core building. The cyclone is detached from any
large-scale, baroclinic structure during its mature phase, with no transfer of PV from the upper-level jet. Conversely, the
upper-level PV streamer is wrapped around the cyclone centre. The PV anomaly at all levels consists in: wet potential
vorticity (WPV) produced diabatically by latent heat release (Eq. 4 in Miglietta et al., 2017) and dry potential vorticity
(DPV) brought by intrusion of stratospheric air into the upper troposphere (their Eq. 3). Levels up to ~ 600 hPa present a
maximum of WPV due to latent heating, while DPV is almost constant up to ~ 400 hPa where it increases sharply, and there
is no real PV tower around the cyclone centre. The features characteristics of tropical cyclones are well marked: warm core
extending up to 800 hPa, perfect symmetry, low-level convergence and upper-level divergence, and strong contrast of $\theta_e$ (~ 8
°C) between the surface and 900 hPa as an evidence of latent heating. The case of October 1996 chosen to represent this
category shows very strong surface fluxes (above 1500 W m$^{-2}$ over large areas) due to strong, persistent winds of orographic





origin bringing cold and dry air for several days prior to the cyclone development, also contributing to destabilize the surface
layer.

Medicanes of the second category also present similarities with tropical cyclones, like deep warm core and symmetrical wind
field, but they lack the diabatism due to latent heat release and the convection is less intense after than before the tropical
transition. The cyclone stays within a large-scale baroclinic environment during its whole lifecycle, with possible barotropic
transfer of PV from the upper-level jet, and the PV streamer evolving into a cut-off low. The tropical-like features are less
evident: weaker warm core, weaker gradient of $\theta_e$ (~ 3−4 °C) between the surface and 900 hPa. Around the cyclone centre,
the potential vorticity forms a PV tower, with weak contrast between the DPV and WPV profiles. As an example, the warm
core of the December 2005 medicane is not due to convective latent heating but to seclusion of warm air by colder air
masses and extends up to 400 hPa. The surface enthalpy fluxes play only a marginal role and take maximum values of 1000
W m$^{-2}$ for a few hours.

In the case of the present study, latent heat release and strong air-sea exchanges at the surface act at building the warm core
anomaly, as seen in Sect. 3.1.3. The surface enthalpy fluxes take intermediate values with maximum above 1500 W m$^{-2}$ for a
few hours on areas with warm SSTs and strong winds downwind of the dry low-level flow from North Africa. No frontal
structures are present after the baroclinic phase. Thermal features characteristic of tropical cyclones are present, like low-
level cold air advection from the south to the east, and warm air advection from the south to the north (Reale and Atlas,
2001), and the gradient of $\theta_e$ between the surface and 900 hPa takes intermediate values of 6−7 °C. There is no upper-level
cut-off, even in the decaying phase of the cyclone, but the wrapping of the PV streamer around the cyclone centre.
Conversely, some typical features are not present: even if there is weak low-level convergence around the cyclone centre, no
divergence is obtained at upper level. The area of maximum latent heat flux within the EF600 area is more controlled by the
SST than by the wind speed (Fig. 13b and 16a, b, and e). No minimum of potential temperature or potential vorticity develop
at 300 hPa close to the cyclone centre during the mature phase, as a marker of the PV anomaly erosion by the convective
activity, and the upper-level PV anomaly never completely detaches from the large scale structure.

The vertical profiles of PV, DPV and WPV (defined as in Miglietta et al., (2017)) averaged on a 100-km radius circle around
the cyclone centre show a minimum of WPV between 700 and 400 hPa during the diabatic phase, and a clear difference
between DPV and WPV at low level (Fig. 18). The DPV is weak up to the mid troposphere and increases sharply above 400
hPa. The WPV anomaly at low levels that develop up to 700 hPa due to convection and latent heating during the transition
phase is increased but reduced in its vertical extent to 800 hPa at the beginning of the diabatic phase (13:00 UTC – see also
Fig. 19e). This is due to a dry air intrusion during the transition and diabatic phase, which is limited downwards to mid
troposphere because of this warm core (Fig. 19a). During the diabatic phase, until 18:00 UTC, the latent heating within the
cyclone core increases the WPV at low levels and erodes the dry and cold ($\theta_e$) air masses up to 650 hPa. Then, the warm core
and WPV anomaly extend upwards (Fig 19b, f), and the DPV anomaly is pushed up to 700 hPa (Fig. 19c, d).



This suggests that the medicane of November 2014 as simulated in this study presents intermediate characteristics, with purely diabatic processes based on latent heat release during its mature phase, but a weaker convective activity than the case of October 1996. Its development phase is triggered by a PV streamer bringing instability at upper level, and baroclinic processes followed by strong convection at sea enhanced and maintained by cold pools due to rain evaporation at low level

or by advection of dry and cold air from North Africa. Conjunction of advection of continental air masses with evaporation under storms has not been identified as leading to tropicalization of Mediterranean cyclones so far, even though it is probably rather ubiquitous, as both are rather widespread phenomena in the Mediterranean Basin. Surface fluxes are strong and contribute to enhance the convection potential till the mature phase of the cyclone. The evaporation is mainly controlled by the SST and by the wind speed during the whole event, while the temperature difference between the SST and the cold air

advected from North Africa during the baroclinic and transition phase play a strong role during its development. The vertical development of the warm core is limited, at the beginning of the diabatic phase, by a dry air intrusion that do no reach the lowest levels of the troposphere. Dry air intrusions have been recognized as common processes in Mediterranean cyclones by Flaounas et al. (2015) but their role in the cyclone lifecycle was not clearly assessed. Here, we suggest that they can act at limiting the extent of the convection at the beginning of the mature phase.

Coupling the atmospheric model with a 3D high-resolution oceanic model shows that the surface cooling susceptible to affect the surface fluxes is too weak in that case to impact the atmospheric destabilization processes at low level. Nevertheless, the effect of the medicane on the oceanic surface layer is probably significant. To better understand the sea surface evolution and the role of coupling, the ocean mixed layer response to the medicane and the mechanisms involved will be investigated in more details in future work.


**Author contributions**. MNB and CLB designed the simulations. MNB performed the simulations. Both authors interpreted the results and wrote the paper.

**Competing interests**. The author declare that they have no conflict of interest.

**Acknowledgments**

This work is a contribution to the HyMeX program (Hydrological cycle in the Mediterranean EXperiment - http://www.hymex.org) through INSU-MISTRALS support. The authors acknowledge the Pôle de Calcul et de Données Marines for the DATARMOR facilities (storage, data access, computational resources). The authors acknowledge the MISTRALS/HyMeX database teams (ESPRI/IPSL and SEDOO/OMP) for their help in accessing to the surface weather



station data. The PSY2V4R4 daily analyses were made available by the Copernicus Marine Environment Monitoring Service (http://marine.copernicus.eu). The authors also thank J.-L. Redelsperger (LOPS) for valuable discussions.

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





## Tables

|       | $U_{10}$ | $\theta$ | SST   | $q$   |
|-------|-------|-------|-------|-------|
| *H+LE* | 0.55  | -0.43 | 0.35  | 0.26  |
| *LE*   | 0.59  | -0.09 | 0.40  | 0.12  |
| *H*    | 0.31  | -0.81 | 0.12  |       |
| $U_{10}$ |       | -0.06 | -0.26 | 0.72  |
| $\theta$ |       |       | 0.01  | -0.12 |
| SST    |       |       |       | -0.27 |

**Table 1:** Spearman's rank correlations between the enthalpy flux, latent and sensible heat flux and related parameters (10 m wind speed $U_{10}$, potential temperature at 10 m $\theta$, SST and humidity at 10 m $q$) at 04:00 UTC on 7 November, from the CPL simulation, on the EF600 area.

|       | $U_{10}$ | $\theta$ | SST   | $q$   |
|-------|-------|-------|-------|-------|
| *H+LE* | 0.66  | -0.17 | 0.30  | 0.52  |
| *LE*   | 0.63  | 0.17  | 0.31  | 0.34  |
| *H*    | 0.42  | -0.70 | 0.14  |       |
| $U_{10}$ |       | 0.06  | -0.31 | 0.85  |
| $\theta$ |       |       | 0.02  | -0.13 |
| SST    |       |       |       | -0.22 |

**Table 2:** Same at Table 1 at 10:00 UTC on 7 November.





|        | $U_{10}$ | $\theta$ | SST   | $q$   |
|--------|----------|----------|-------|-------|
| *H+LE* | 0.43     | -0.05    | 0.33  | 0.29  |
| *LE*   | 0.26     | 0.28     | 0.48  | 0.05  |
| *H*    | 0.50     | -0.73    | -0.22 |       |
| $U_{10}$ |        | -0.09    | -0.45 | 0.93  |
| $\theta$ |        |          | 0.45  | -0.15 |
| SST    |          |          |       | -0.44 |

**Table 3:** Same at Table 1 at 17:00 UTC on 7 November.



**Figures**



**Figure 1:** Map of large-scale domain D1, with the domain D2 indicated by the solid-line frame and the area of interest (AI) indicated by the dashed-line frame.







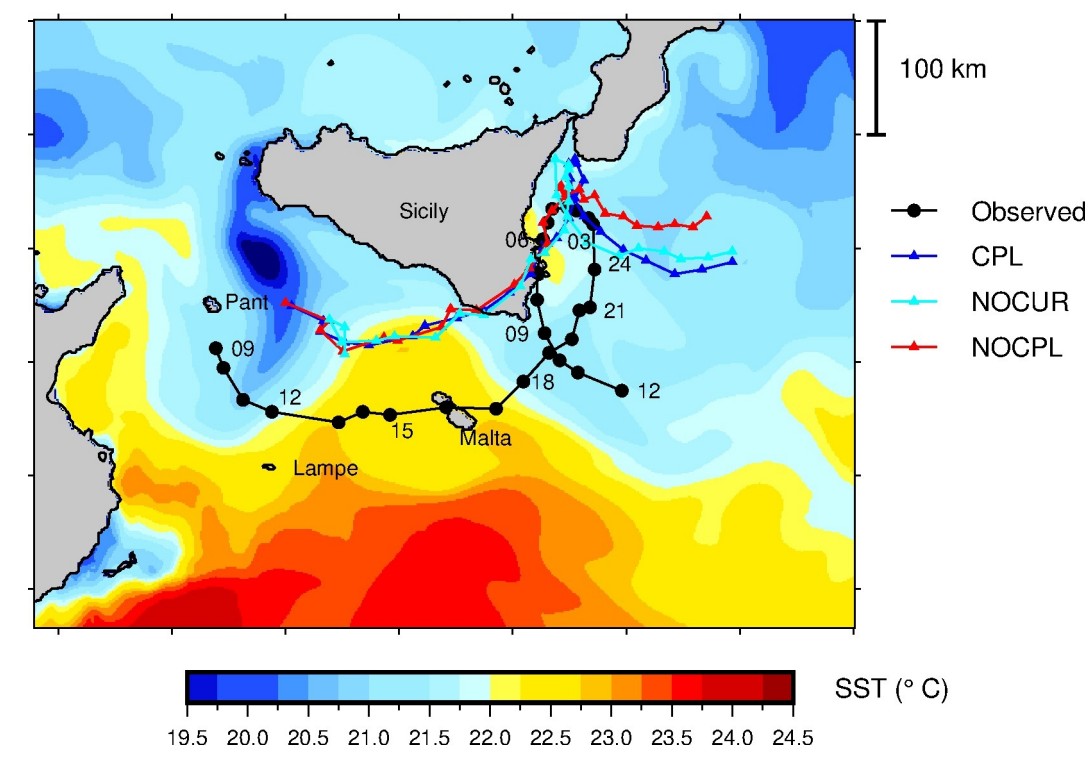




**Figure 2:** Comparison of the simulated trajectories (triangles) of the non-coupled run (NOCPL, red), coupled run with SST only (NOCUR, cyan) and fully coupled run (CPL, blue) with the best track (black closed circles) based on observations as in (Cioni et al., 2018). The position is shown every hour with time labels every 3 h, starting at 09:00 UTC on 7 November until 12:00 UTC on 8 November. In colours, initial Sea Surface Temperature (SST, °C) at 01:00 UTC on 7 November.





**Figure 3:** Time series of the maximum of the 10 m wind speed, and 10 m wind averaged over a 100 km radius around the cyclone centre (a) and minimum sea-level pressure (b) as obtained in the different simulations on the 7 November and 8 November until 12:00 UTC. The thin red line in (a) indicates the 18 m s$^{-1}$ wind speed threshold. The observations of SLP in Linosa (black plain circles) are shown for 820 comparison in (b) – see text.




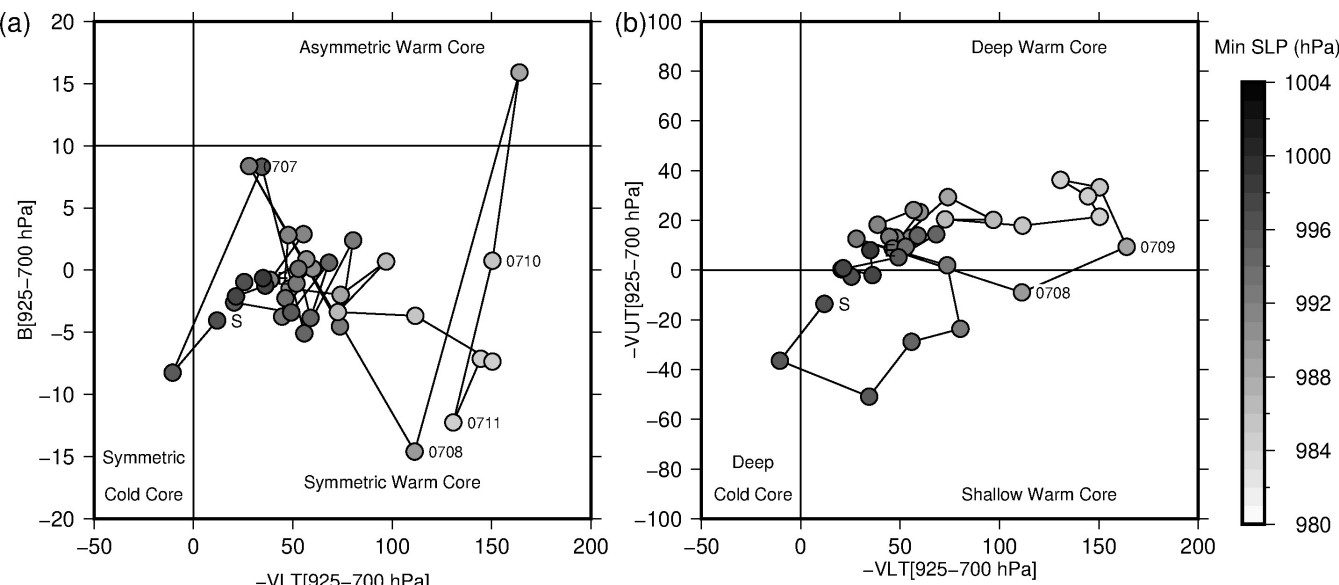

**Figure 4:** Hourly phase diagram of the NOCPL simulated cyclone from 01:00 UTC on 7 November (S) till 12:00 UTC on 8 November, with low-tropospheric thickness asymmetry inside the cyclone ($B$) with respect to low-tropospheric thermal wind ($- V_{LT}$), (a) and upper-tropospheric thermal wind ($-V_{UT}$) with respect to low-tropospheric thermal wind (b).






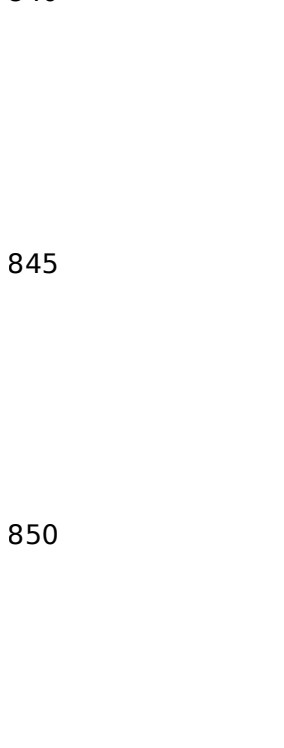

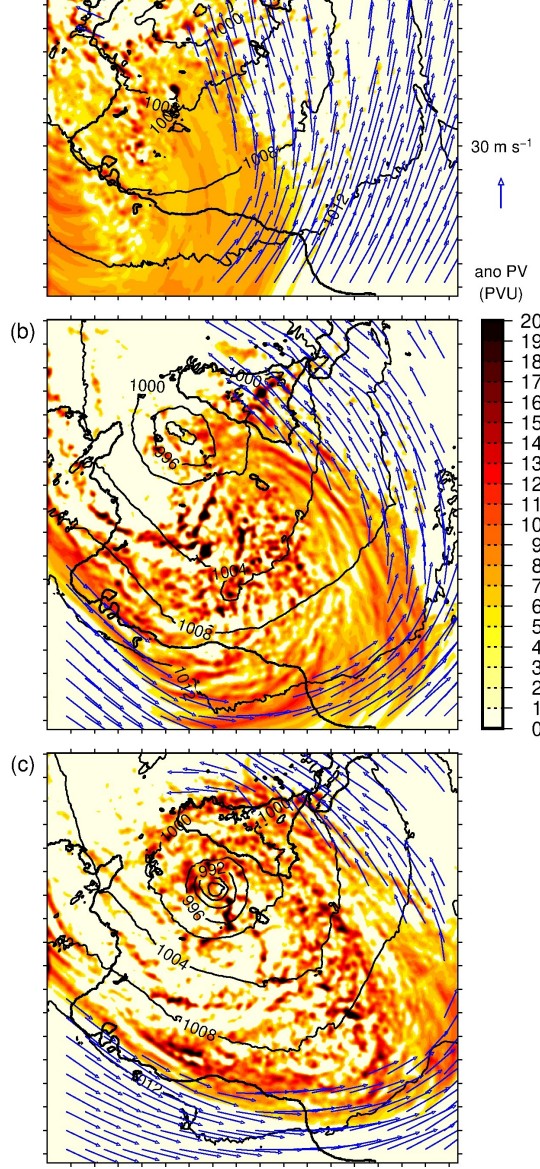

**Figure 5:** Maps of potential vorticity anomalies at 300 hPa (color scale), horizontal wind at 300 hPa above 30 m s$^{-1}$ (blue vectors) and SLP (black contours) at 02:00 UTC (a), 08:30 UTC (b) and 12:00 UTC (c) on 7 November.







**Figure 6:** Map of the potential temperature (°C) at 10 m (color scale) with 10 m wind (black vectors), horizontal convergence rate above 1 $10^{-3}$ m s$^{-2}$ (white contours), instantaneous rain rate above 5 mm h$^{-1}$ (green contour), PV anomaly at 300 hPa above 2 PVU (dark red contour), and SLP (black contours) at 04:00 UTC on 7 November (a), and vertical cross-section of potential temperature (color scale), tangential wind (black vectors, the vertical component is amplified by a factor 20), potential vorticity anomaly (white contour at 5 PVU) and instantaneous rain rate (green bars). The black dashed line in (a) corresponds to the cross section (b). The grey star indicates the position of the SLP minima.






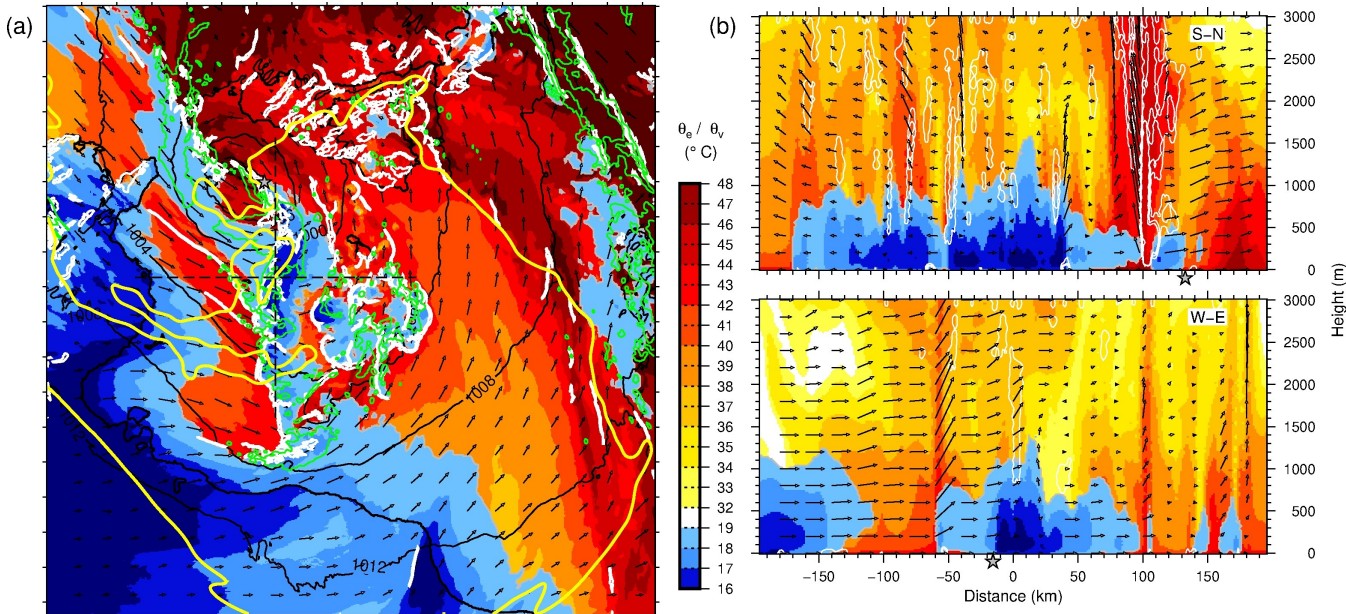

**Figure 7:** Map of equivalent potential temperature (warm colors) and virtual potential temperature below 19 °C (blue shades), horizontal convergence rate above 1 $10^{-3}$ m s$^{-2}$ (white contours), instantaneous rain rate above 5 mm h$^{-1}$ (green contour), PV anomaly at 300 hPa above 2 PVU (yellow contour), and SLP (black contours) at 08:30 UTC on 7 November (a), and vertical cross-sections of equivalent

potential temperature and virtual potential temperature (colour scale), tangential wind (black vectors, the vertical component is amplified by a factor 20), potential vorticity anomaly (white contour at 5 PVU) along south-north (b) and west-east (c) transects. The black dashed lines in (a) indicate the S-N and W-E transects. The grey star indicates the position of the SLP minima.




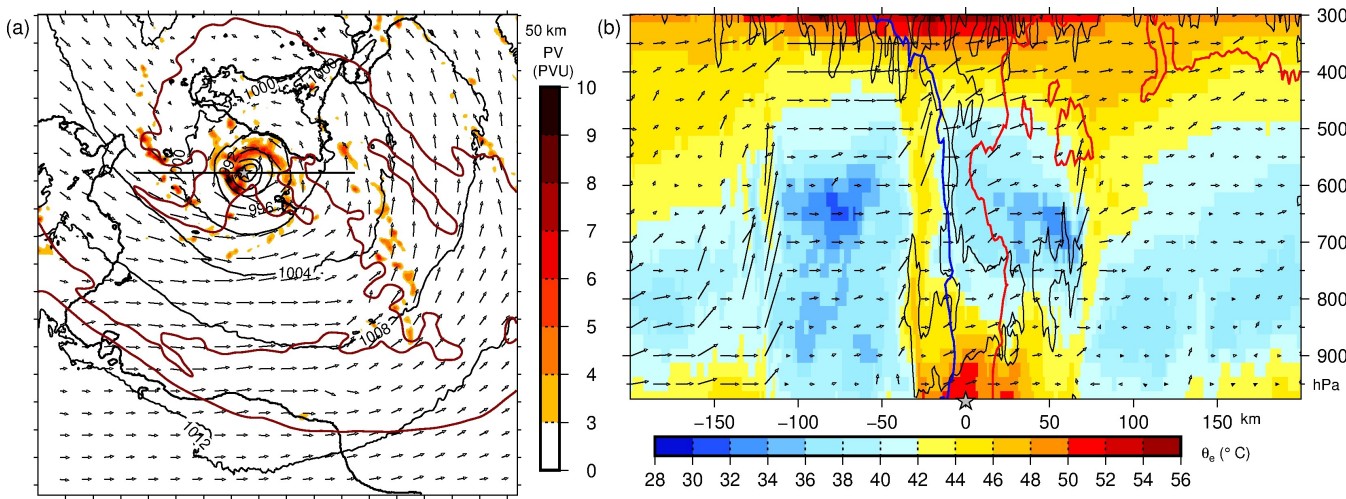

**Figure 8:** Map of potential vorticity anomalies at 700 hPa above 3 PVU (color scale) and 300 hPa (dark red contours at 5 PVU), and 10 m

horizontal wind (black vectors), and SLP (black contours) (a), and vertical cross-section of equivalent potential temperature (color scale),

tangential wind (black vectors, the vertical component is amplified by a factor 20), potential vorticity anomaly (black contour at 5 PVU),

and meridional wind speed below -8 m s$^{-1}$ and above 8 m s$^{-1}$ (blue and red contours respectively) – (b) at 13:00 UTC on 7 November. The

black dashed line in (a) corresponds to the cross section (b). The grey star indicates the position of the SLP minima.







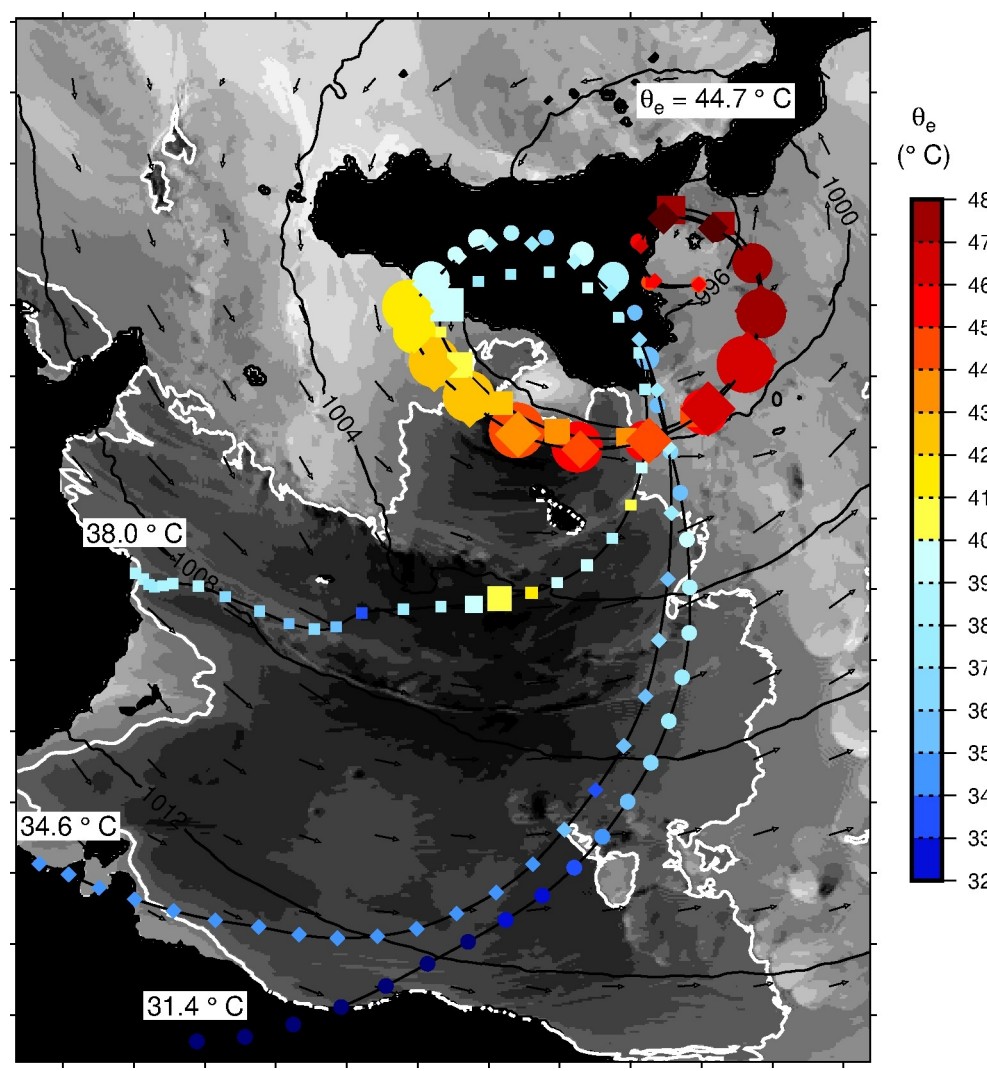

**Figure 9:** Map of the backtrajectories of air parcels arriving south of the cyclone centre at 23:00 UTC on 7 November, 1500 m above sea level, at 3 different levels (circles, squares and diamonds). The colour scale indicates the equivalent potential temperature (°C) and the size of the symbol is inversely proportional to altitude between 0 and 1000 m. Are also shown the values of the final equivalent potential temperature, of the initial equivalent potential temperatures, the wind field at 900 hPa (black vectors), and the surface enthalpy flux (grey shades) with a threshold at 600 W m$^{-2}$ (white contour) at 15:30 UTC when the particles arrive at sea south of Sicily.





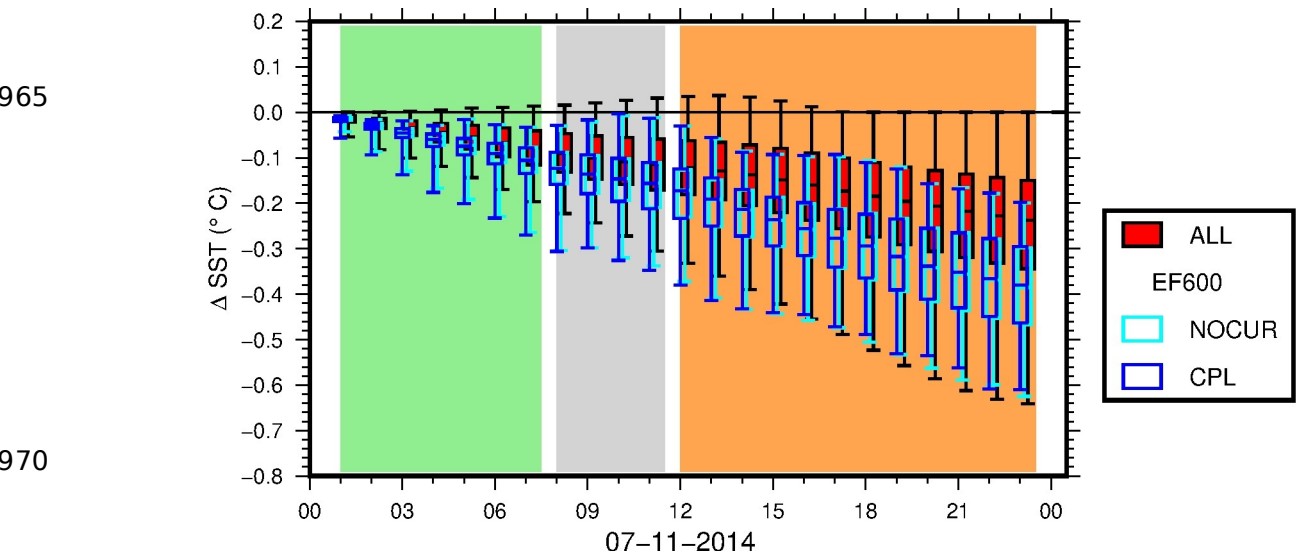

**Figure 10:** Time series of the median differences between the SST in the CPL and NOCPL simulations, on the whole domain (red) and on the EF600 area (blue, see text for definition), on the 7 November. The boxes indicates the 25 and 75% quantiles and the whiskers the 5 and 95% quantiles. Are also shown the differences between the NOCUR and NOCPL simulations (cyan). The background shading indicates the baroclinic (green), transition (grey) and diabatic (orange) phases. Some of the boxes have been slightly horizontally shifted for clarity.



**Figure 11:** Time series of the mean values and standard deviation (error bars) of the total turbulent heat flux (blue), latent (cyan) and sensible heat flux (red) in the CPL (open circles) and NOCPL (triangles) simulations (a) and of the mean difference between CPL and NOCPL turbulent fluxes (open circles, same colour code) and between NOCUR and NOCPL turbulent fluxes, in percent relative to the NOCPL values (b) on the EF600 area. The background shading indicates the different phases.








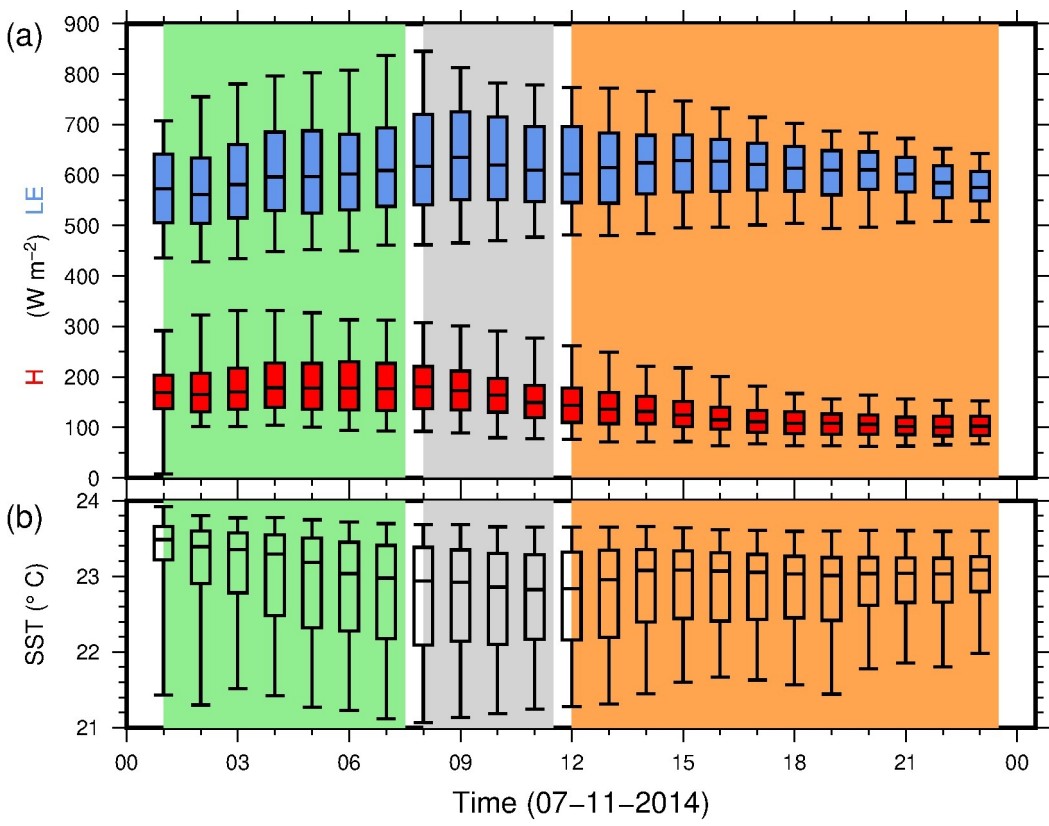

**Figure 12:** Time series of the median values of latent (blue) and sensible heat fluxes (red, a) and of SST (b) on the EF600 area (see text) on the 7 November. The boxes corresponds to the 25 and 75% quantiles, the whiskers to the 5 and 95% quantiles. The background shading indicates the different phases.







**Figure 13:** Time series of Spearman's rank-order correlation $r_s$ between the latent heat flux *LE* and 10 m wind speed (green), potential temperature at 10 m (red), SST (blue) and specific humidity at 2 m (cyan) on the whole domain (a) and EF600 area (b), in the CPL simulation. The background shading indicates the different phases.







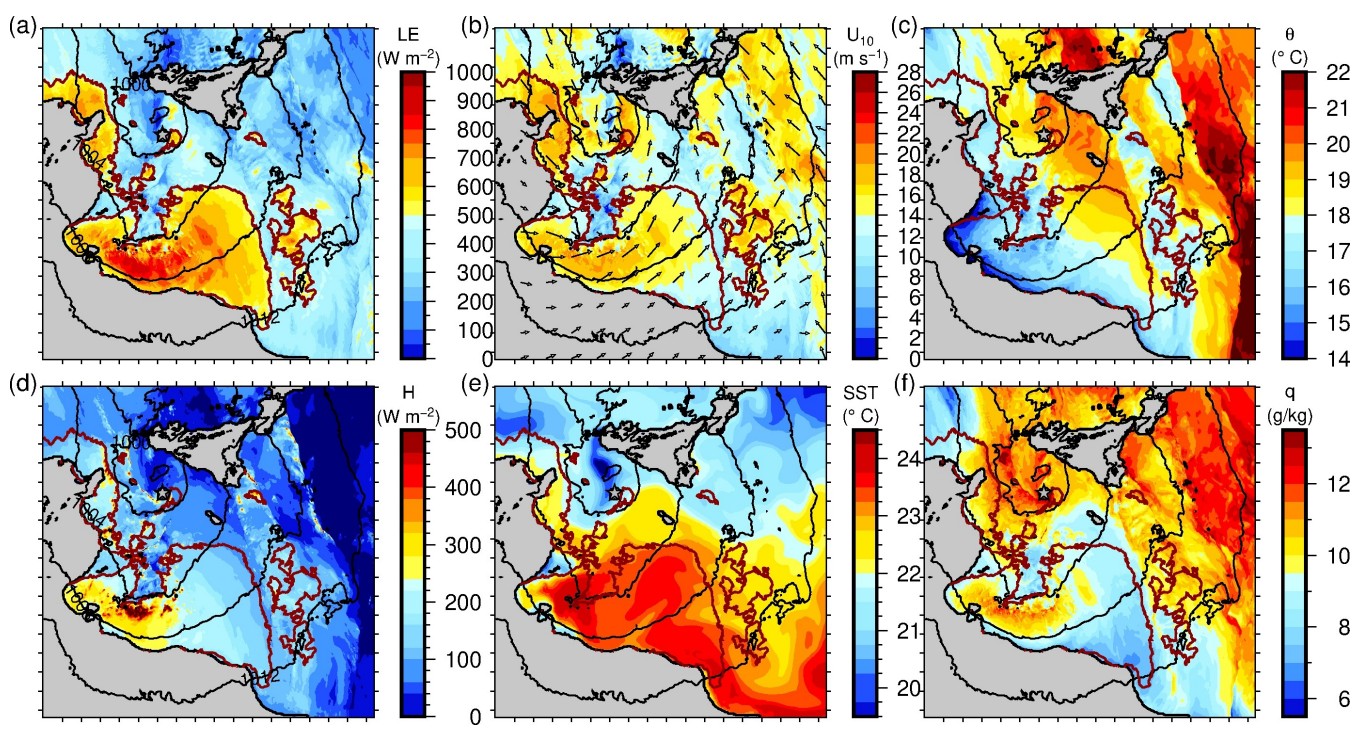

**Figure 14:** Maps of the total turbulent heat fluxes *LE* (a), *H* (b), the 10 m wind $U_{10}$ (c), the 10 m potential temperature (d), the SST (e) and the specific humidity at 2 m (f) at 04:00 UTC on 7 November, in the CPL simulation.








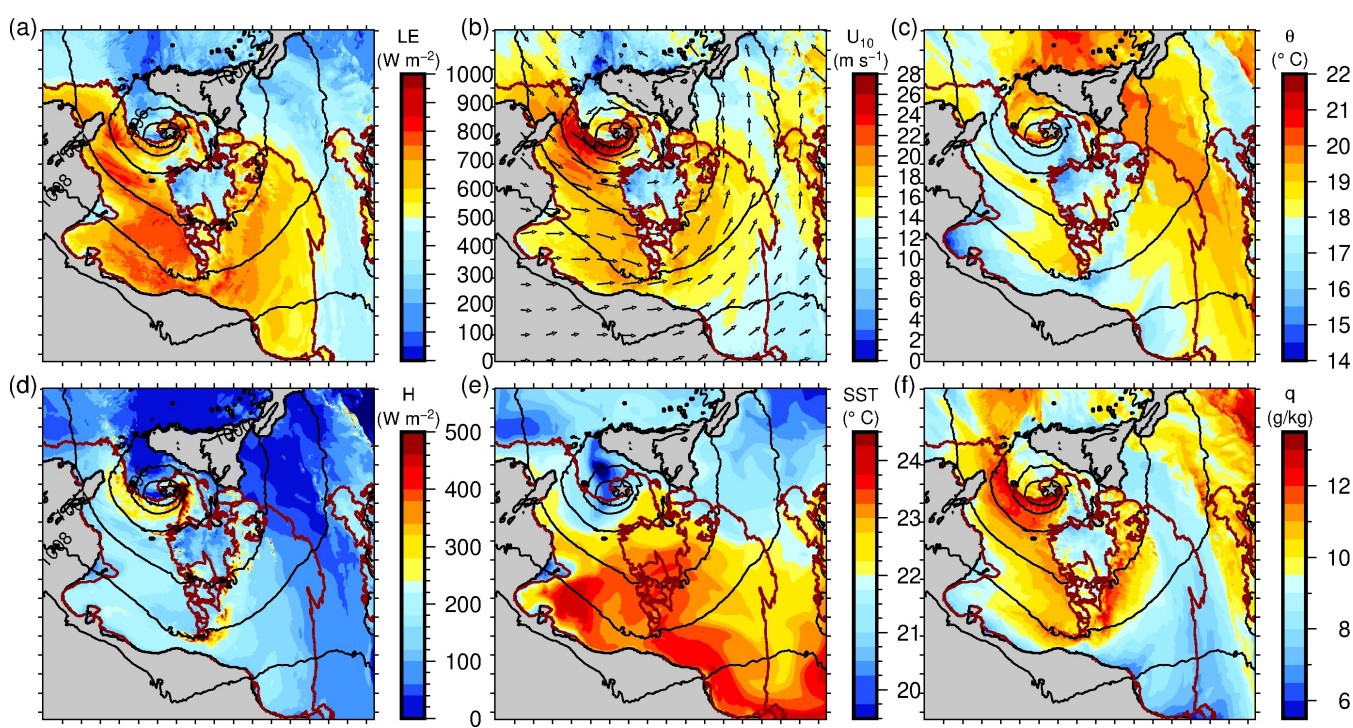

**Figure 15:** Same as Figure 14 but at 10:00 UTC on 7 November.







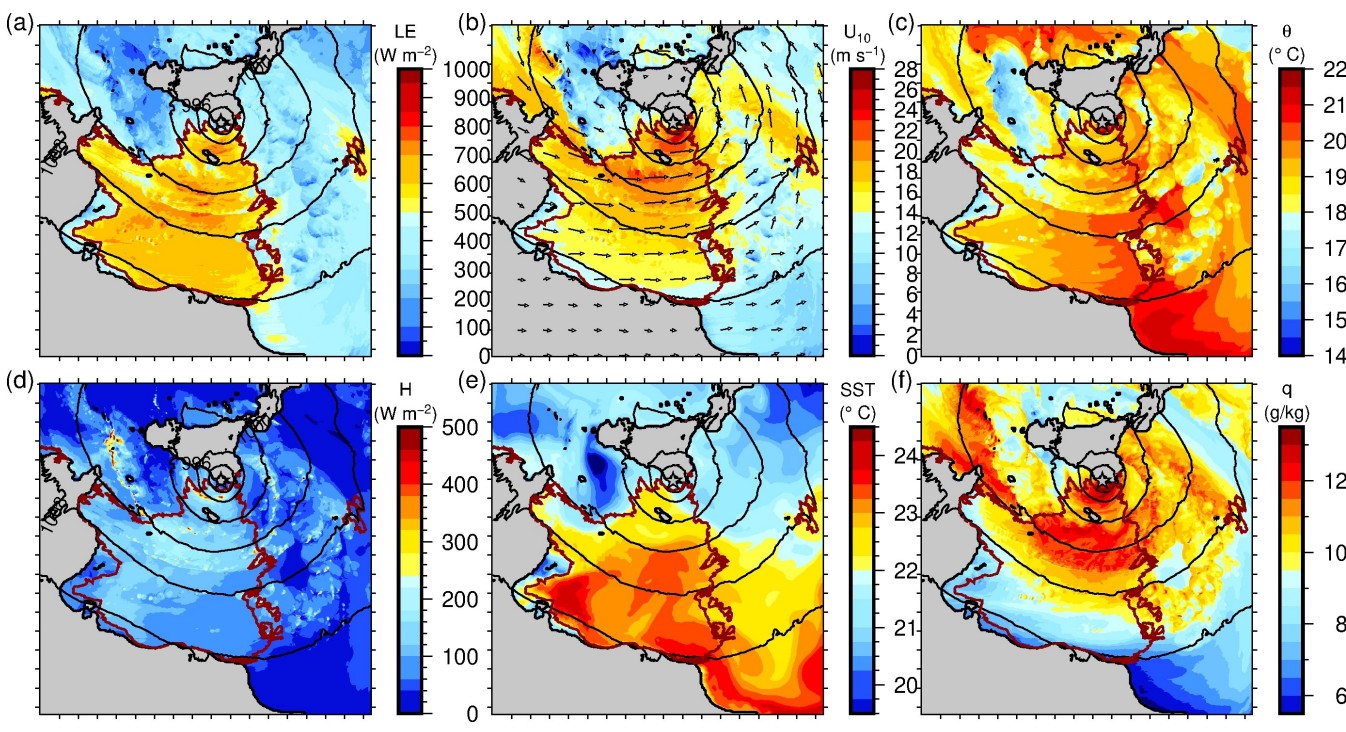

**Figure 16:** Same as Figure 14 but at 17:00 UTC on 7 November.








**Figure 17:** Same as Figure 13 but between the sensible heat flux *H* and 10 m wind speed (green), potential temperature at 10 m (red), and SST (blue) on the whole domain (a) and EF600 area (b), in the CPL simulation.




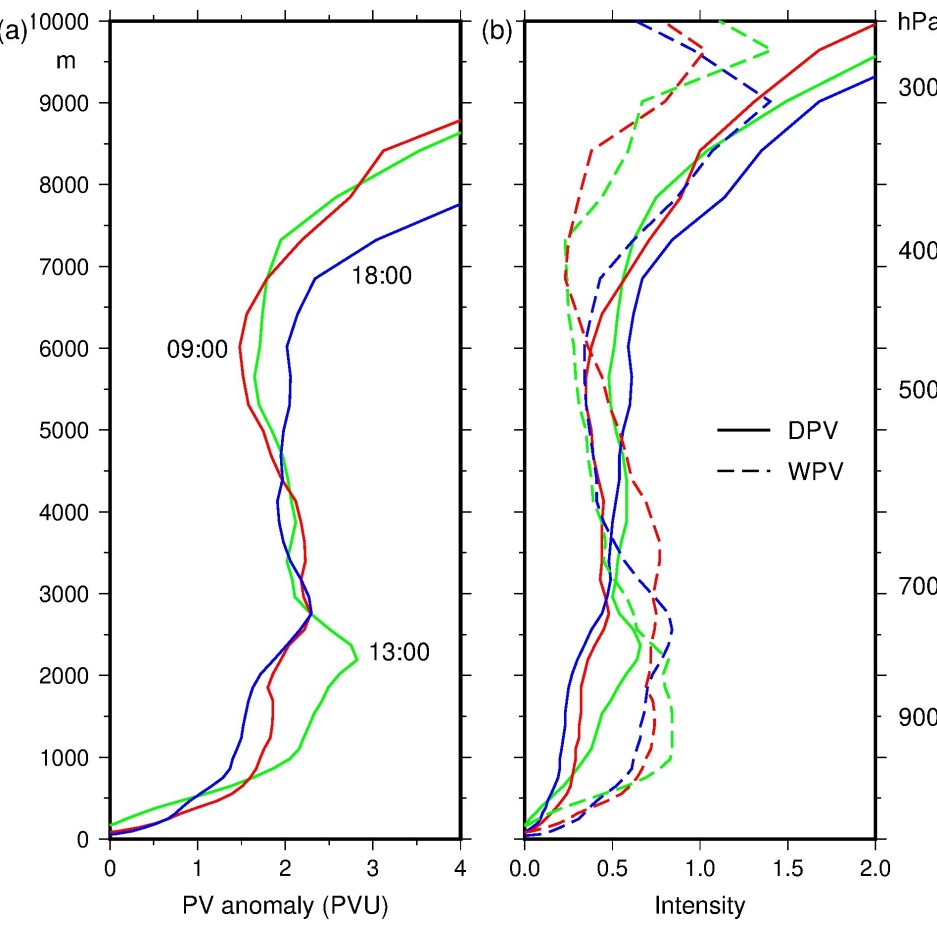

**Figure 18:** Vertical profiles of PV (a), and DPV and WPV (b) averaged within a 100-km radius circle around the cyclone centre at 09:00
(red), 13:00 (green) and 18:00 UTC (blue) on 7 November, in the CPL simulation.







**Figure 19:** Vertical cross-sections of equivalent potential temperature $\theta_e$ (°C, colour scale) and relative humidity (%, isolines), (a,b), DPV (intensity), (c,d) and WPV (intensity), (e,f) on a west-est transect across the cyclone centre, at 13:00 (a,c,e) and 18:00 UTC (b,d,f) on 7 November, in the CPL simulation. The black contours in (c) to (f) correspond to intensities (as defined in Miglietta et al., 2017) 1 and 3.