# Peer review of "Surface processes in the 7 November 2014 medicane from air-sea coupled high-resolution numerical modelling"

_Atmospheric Chemistry and Physics, 2019_

## Referee Comment (RC1) · Anonymous Referee #1 · 21 Nov 2019

RECOMMENDATION: Major revisions

The Medicane of November 2014 is analyzed here using a coupled modelling approach, focusing on the role of surface processes for the development of the cyclone. This part of the analysis is very detailed and provides a clear investigation of the role of different surface parameters involved in the evolution of the cyclone. Also, results show that the coupling with the ocean model appears not necessary for a proper simulation of this Medicane.

Results are interesting and worth of publication, in particular Section 3.2 and 4 are full of interesting insights in the mechanism of development of the cyclone in terms of air-

sea interaction processes. Also, bibliography is detailed and provides a very updated description of the state-of-the-art in the field.

In contrast, I have some concerns on the discussion of the NOCPL simulation (Section 3.1), for which I ask to reconsider, at least in part, the analysis provided in the paper. Thus, my recommendation is for a major revision.

Major points:

- The distinction among the three different phases in the cyclone lifetime is subjectively defined, thus not completely convincing. I recommend you find an objective way, e.g. based on the Hart (2003) diagram, or on the methodology discussed in Fita and Flaounas (2018).

- The discussion is Section 3.1 is not convincing in many points (see also minor points in 3.1.1 and 3.1.2):

L280-285: the bands of colder air may not be due to the evaporation of precipitation. In particular, the one to the left may be due to cold-air advection, which secludes the cyclone warm core, turning around the center within the cyclonic circulation, as suggested by Fig. 6a. To clarify this aspect, one should perform numerical simulations without the evaporation of precipitation: this would demonstrate whether the cold air is due to a long-range transport toward the center of the cyclone or really to the evaporation of precipitation.

L286-288: I understand that your focus is on surface fields, but you should also consider what happens in the levels immediately above (e.g., 950 hPa, 900 hPa) to support your considerations and better identify the origin of the different air masses and thermal gradients. The vertical extension of the cold air masses in North African may be limited to a few meters, so you should demonstrate more clearly that the "advection of cold and dry air . . . from the Tunisian and Libyan continental surface . . . (L296-297, L448-449)" is relevant. For example, you can use backtrajectories, going earlier in time

than those shown in Fig. 9, to clearly illustrate the origin of the air parcels.

Minor points: Lines 9-12: "The deepening . . . of the cyclone is due . . . then to low-level convergence and uplift of conditionally unstable air masses by cold pools, resulting either from rain evaporation or from advection of continental air masses from North Africa.": the deepening can be due to WISHE mechanism and/or to baroclinic processes, the way precipitation is generated is related to convergence and uplift by cold pools, so the sentence should be reformulated.

Line 14: . . . due to a sea surface cooling . . .

Line 27: at least up to the mid troposphere . . .: really, even in the version of Hart diagram modified by Picornell et al. (2014) you used, the vertical structure is analyzed also in the upper troposphere, up to 400 hPa.

Line 33: note that the 26°C threshold of tropical cyclones cannot be applied to tropical cyclones developing from tropical transition processes (see McTaggart-Cowen et al., 2015).

Line 41, 274: I image you are referring to lee cyclones (see Tibaldi et al., 1990), not to lee waves. Lee cyclones are not only the effect of a wave, but theory provides a comprehensive way to describe the process of cyclogenesis.

Line 119: . . . its initiation as a baroclinic storm . . .

Section 2.1: I understand that 19 Figures are a lot, but please consider the possibility to include additional 1-2 figures to make easier to understand the results in this Section, or refer to Figures published in previous papers.

Line 144: please refer to Fig. 1 to help following the text.

Line 170: Is convection treated explicitly? Please add this piece of information.

Line 171: grid spacing instead of resolution.

[Figure]

Line 189: what is an ORCA grid?

Section 2.2.3: what is the rationale behind the choice of this domain extension? Have you tried with different domains? Apparently, only a small domain extension is favorable for a proper simulation of the track - cfr. Cioni et al. (2018) and your tracks with those in Carriò et al. (2017) and Pytharoulis et al. (2018). Please, comment on this.

Figure 3a: it would be helpful to add at least the maximum wind recorded in Lampedusa, Malta, Pantelleria in panel a.

L274: ... by the lee cyclone induced by the North African relief ...

L279-280: note that it is not the convergence between SE flow and S-SW flow responsible for the most intense precipitation at sea in Fig. 6b, but the one associated with the low-level southwesterlies and the northwesterly flow behind.

L284: northwestwards?

L285: deep convection in ...

L292: The low-level virtual ...

L300: in the southeasterly low-level flow ... (in Fig. 7b, the wind component seems from the south).

L302-305: did you check these points in the simulation? May you be more quantitative?

L306: the high CAPE is not obtained by extracting heat and moisture! The surface fluxes modify the low-level features, determining an environment more favorable to instability (i.e., CAPE increased).

L316-318: again, this sentence should be less qualitative in order to be more convincing.

L323: Mazza et al. (2017) does not refer to the December 2005 Medicane.

L331: you really show the role of air-sea interaction processes; the role of diabatic

processes may be inferred from theory, you do not show that it occurs in this case.

L342-343: I suggest to change "the evolution of the SST" with "sea currents".

L360: an important conclusion you could mention here is the negligible role of currents.

L384: the gradient of wind speed . . . between which levels?

L385: I suggest to reformulate as "humidity at saturation with temperature equal to SST".

L394-398: this part is already discussed and can be omitted.

L410-411, L421: This is partly due to the conditional sampling . . .: explain better.

L424-425, L439: The sentences "The time evolution of the distributions of LE and the SST are opposite to each other" and "parameters controlling LE (and evaporation) are the SST . . . with very strong positive correlations" do not contradict somehow each other?

L442: please change "globally" into "positive".

L442-443: this sentence is not clear; my interpretation is that you should rearrange in something like: "The fact that r is low in the whole domain, and higher in EF600 suggests that strong evaporation controls specific humidity and temperature".

L448-449: see major point.

L485: in contrast . . .

L485-486: again, see major point.

L531: . . .lack diabatism . . .: really, the second case discussed in Miglietta and Rotunno (2019) does not lack diabatism, but contains both diabatism and baroclinic processes. Please clarify this point here and later in the discussion (L561-564 should be changed, as the Medicane apparently belongs to the second category of Medicanes).

[Figure]

L555: This is consistent with the observations in Miglietta et al. (2013) and Dafis et al. (2018).

Figure 6 is very confusing: the coastline can be hardly identified; latitude and longitude are not reported; the extension of the cross section in b) is not clear; finally, the same color scale should be used in both panels. Similar considerations apply to Figure 7.

Figure 9: how long do the backtrajectories go back in time?

Figure 10: the cyan and blue columns are difficult to distinguish.

Figure 11: the triangles are difficult to identify.

REFERENCES:

Dafis, S., Rysman, J.-F., Claud, C. and Flaounas, E. (2018) Remote sensing of deep convection within a tropical-like cyclone over the Mediterranean Sea. Atmospheric Science Letters, 19(6), e823.

McTaggart-Cowan, R., Davies, E.L., Fairman, J.G., Jr., Galarneau, T.J., Jr. and Schultz, D.M. (2015) Revisiting the 26.5åŮęC sea surface temperature threshold for tropical cyclone development. Bulletin of the American Meteorological Society, 96, 1929–1943.

Tibaldi, S., Buzzi, A., and Speranza, A.: Orographic cyclogenesis, in: The Palmen Memorial Volume, edited by: Newton, C. and Holopainen, E. O., Am. Meteorol. Soc., Boston, 107–127,1990.

---

## Referee Comment (RC2) · Emmanouil Flaounas (Referee) · 16 Dec 2019

I read with great interest the study by Bouin and Lebeaupin Brossier, entitled "Surface processes in the 7 November 2014 medicane from air-sea coupled high-resolution numerical modelling". The results have been reached using adequate methods and a plethora of diagnostics. They are valuable for the Mediterranean cyclones community and therefore I am certainly in favour to see this study published. Nevertheless, I have several major concerns on the analysis of the results.

Major comments 1) I find that the technical language used to describe atmospheric dynamics in several parts of the text comes at the cost of a clear interpretation of

cyclone dynamics. Here are several examples from the abstract and introduction. I suggest a more careful revision of technical language throughout the text.

line 10: "Tropicalization". I am aware of "tropical transition" but I am afraid I am not familiar with this term. What is its physical/dynamical content? If this term has been used before, then its physical content could be very briefly explained alongside with a reference, if not then, maybe it should be defined in the text.

line 10: ".. then to low-level convergence and uplift of conditionally unstable air masses by cold pools..". Is it meant that "tropicalization" is a product of jet crossing and latent heat release? Then from the perspective of cyclone dynamics maybe it should be mentioned that both baroclinic and diabatic forcings act in synergy to deepen the cyclone.

line 12: "...feeding the latent heat release during the mature phase of the medicane..." do you mean that specific conditions favour convection, in turn leading to high latent heat release? Such phrases were found in several other parts of the manuscript.

line 40: A PV streamer is filament of high PV flow it may not be always regarded as equivalent to a trough.

line 40: "potential vorticity (PV) from higher-latitude regions.." you mean intruding stratospheric air characterized by high PV values?

lines 40-43: I am not sure how these lines help us understand the contribution of different processes in medicanes development.

line 46: what do you refer by "upper-level thermodynamics". What is meant by "limited", maybe you mean "inadequate"?

Line 49: "behaviour", "extension", "..various characteristics" please be specific by rephrasing or attributing a physical content to each term. Please also note that Fita and Flaounas (2018) that warm seclusion may be diagnosed as a warm core symmetric system even if convection is weak. This is opposed to tropical cyclone dynamics.

Line 53: "PV transfer from above" I am not sure what is meant here. Do you suggest that a cyclone was intensified due to the formation of a PV tower after aligning air of stratospheric origin with diabatically produced PV in the lower to mid troposphere?

Line 56: As formulated here it seems that the PV streamer and the upper level Jet are two different atmospheric features.

Line 60: "large diversity of characteristics and behaviours". Please provide physical content to these words.

Line 61-62: I am not sure I understand this phrase. In main lines PV anomalies in a given location can originate from advection (adiabatic process) or from momentum/heat exchanges with the environment (diabatic). Both contribute to the formation of a PV tower and thus both provide a combined baroclinic or diabatic forcing to cyclonic circulation.

Lines 74-77: This phrase is long and difficult to understand.

Line 76: please note that atmospheric processes (e.g. convection, or baroclinic instability) are expected to form, or sustain a vortex/cyclonic circulation not enthalpy or surface thermodynamical variables. What is meant here is that surface processes enhance convection and thus a diabatic forcing to cyclonic circulation?

Lines 82-83: "WISHE-type vs baroclinicity". Please rephrase. Baroclinicity is not a mechanism is an atmospheric state.

Lines 83-84: It is quite awkward to characterize a PV-streamer as "instability aloft". Do you mean that stability decreases within the atmospheric column under the PV-streamer?

Lines 85-86: I am not sure I follow the rational here. Fita and Flaounas, (2018) present a detailed analysis of the implicated dynamics in the development of the december 2005 case. Both a PV-streamer and a cut-off low were involved (the former evolved to the latter). Both features correspond to a PV anomaly of statospheric origin (cut-off or

streamer) that provide a baroclinic forcing to the cyclone. I would say that the difference comes from the extent that baroclinic or diabatic forcing (basically latent heat release due to convection) prevails in provoking cyclonic circulation.

Lines 82 & 87: "WISHE-type" and "WISHE-like". Please be consistent on the terminology. If the mechanism explained here diverges from the original WISHE, please be more specific.

Lines 87-88: I do not understand how warm seclusion "replaces" the "heat and moisture to latent heating through a WISHE-like mechanism". Is it meant here that the low-level warm core in the first case is due to convection and latent heat release, while in the second case it is due to warm seclusion?

Line 90: Would you agree that the "Jet-crossing" mechanism refers to a baroclinic forcing to the cyclone? E.g. Jet-crossing could deepen a cyclone even in the absence of any diabatic processes. This seems to be similar to the upper tropospheric forcing "D$\Phi$" on the medicane case of 2005 in Fita and Flaounas (2018), their Figure 7.

Line 98: Intensity may change due to processes such as convection, not surface heat fluxes. A warmer SST would suggest enhanced convection or at least, favour latent heat release.

Line 108: Please also note that the impact of coupling has limited effect on "all intense Mediterranean cyclones". This was shown in a simulation ensemble by Flaounas et al., (2018). This could reinforce the argument that known medicane cases are not expected to have an exceptional sensitivity to air-sea coupling than the rest of the intense cyclones.

2) Medicanes definition and relevance with this case: A major drawback in the field is the lack of a physical definition for medicanes. I am not sure that it has ever been shown that there is a "perfect vertical alignment between the sea level pressure (SLP) minimum". I could argue that in several papers there is no such alignment observed,

especially when the "eye" is fully developed. In line 30 how is radius defined? I suggest a more careful opening paragraph mentioning that knowledge comes for a very limited number of cases qualified as medicanes where no objective criteria have been used other than some arbitrary visual characteristics of cloud coverage. After all, several papers show that medicanes present a high variability of dynamical structures (e.g. Miglietta et al., 2017).

In fact, from a climatological context, medicanes are not expected to clearly stand out in terms of dynamics from several other cases that were not "qualified" as medicanes. This seems to be one of the results for several of the metrics used by Tous ad Romero (2013) and Flaounas et al., (2015) in line 37.

3) I believe that cyclone dynamics in section 3 are not convincingly addressed (this also concerns section 5). It seems to me that this is also partly due to how technical language is used (comment #1).

In summary a PV streamer intrudes the Mediterranean, it provokes cyclogenesis and wraps around the cyclone centre until it evolves to a cut-off at 300 hPa (this is not very clear due to the domain size in figure 5). This scenario seems to be fairly similar to the 2005 medicane, analysed by Fita and Flaounas (2018) and Miglietta and Rotunno (2019), but also to several other cases.

In the "transitional" phase (Fig. 5b), the main body of the PV anomaly is dislocated with respect to the cyclone centre and overall I would say that this example complies with the classic paradigm of baroclinic instability in Hoskins et al., (1985), as illustrated from a PV perspective. What do you refer to with "no trace of baroclinicity" in line 265? With such high PV anomalies in the upper troposphere, the potential temperature isosurfaces should slope accordingly, creating a barcolinic environment to the proximity of the cyclone.

The analysis in 3.1 is based on the PV-streamer evolution, which mainly refers to adia-batically advected stratospheric air masses. Why do you refer to the last phase of the

section as "diabatic"? How can rainfall be indicative of diabatic vs baroclinic forcing to the cyclone intensification? Actually the presence of high PV anomaly in the upper troposphere suggests a considerable forcing to the surface cyclone.

Figures 6 and 7 are overcharged and not consistent in the plotted fields. It is also quite odd to use potential temperature at surface level. If this is a model diagnostic (e.g. as 2-meter temperature) rather than a prognostic variable, then it is not used by the model to resolve dynamics. This would create some inconveniences in the analysis. Furthermore, the field is strongly influenced by land-sea transitions. I suggest you use 850 hPa for low level baroclinic structures. Finally, the two overlaid temperature fields in Fig. 7 makes interpretation of dynamics rather difficult.

Lines 274-275 "baroclinic processes are responsible for the heavy precipitation" I am not sure I understand this phrase. Are you suggesting that large scale, quasi-geostorphic forced ascent provokes rainfall?

Section 3.1.3: I do not really understand the term "diabatism". If you refer to the mechanism of tropical transition (Davis and Bosart, 2004) then I guess that you suggest that an initially baroclinic system is now maintained due to convection (as stated in line 316). However there is still an upper PV anomaly due to the streamer. How much does it contribute to the surface cyclonic circulation (also evident in the PV profiles of Fig. 18)? Does the absence of divergence in the upper troposphere suggest that the dynamics are different from the ones expected by tropical cyclones (lines 313)?

It seems that the trajectories "confirm that the warm core of the medicane is actually due to diabatic processes. (lines 330-331)". How do high $\theta$e values (regardless the origin of these values) assure that the cyclone is maintained due to diabatic forcing? Equivalent potential temperature ($\theta$e) should remain constant if condensation takes place. Here, air masses experience a dramatic increase of $\theta$e. Does this reflect an increase in heat fluxes or moisture? Furthermore, how often do the air masses hit the ground and what is the time interval used to calculate the trajectories? Could you

comment on how the method used to calculate the trajectories assures that the "same" air mass is followed in time (especially given the sudden change of altitude after leaving the Sicilian coast)?

In fact, factor separation technique in Carrio et al., (2017) shows that the development of this cyclone is due to synergy between upper tropospheric forcing and latent heat release (their figure 14). Could you compare and discuss with respect to their results?

4) I suggest to reorganise the manuscript structure to make it more attractive to the reader. Section 4 is very intensive, the importance of the results is eclipsed by a plethora of diagnostics and variables and seems quite detached from section 3. I would suggest to the authors to make a clearer and more refined presentation of the objectives in the introduction and merge the sections 3 and 4 by organising the document sections according to cyclone phases. In fact, section 5 seems to include quite interesting analysis that would be more adequate for section 3.

Minor comments: Abstract: One important message is that intense air-sea interactions have not a strong effect on the cyclone development. Please further highlight your main results, I am not sure about what message we get from lines 16-21.

Line 131: It seems from Di Muzio et al., (2019) that the occurrence of Qendresa was well predicted by ECMWF with a lead time of 7-8 days in contrast to most medicanes, where predictability seems to be possible only in erlier lead times. Could you please comment, or be more specific, on the predictability of this case?

Line 213: Typically trajectories refer to air masses and tracks to cyclones. What "best-track" refers to and what kind of observations were used?

Line 226-227: This is contradictory statement please rephrase. In fact, the northward displacement of the track only allows to confirm an agreement of the deepening phase of the cyclone between observations and model.

Line 239: I do not follow the reasoning of using 100 km radius, been "fitted to the radius

[Figure]

of maximum wind". The 100 km radius used is too small, compared to the size of the cyclonic circulation. This is quite clear from the wind field in e.g. Fig. 6. Therefore it seems quite difficult for the diagnostic to capture asymmetries due to frontal structures, which in turn develop in higher length scales. Could you please comment on this? Furthermore Di Muzio et al., 2019 used the operational analyses and same set-up for cyclone phase diagrams and argues that Qendresa presented no warm-core in the upper troposphere. Could you comment on the sensitivity of phase diagrams in your case?

Figure 4 Legends in all axes seem to mention the same pressure layers.

Line 360: Could you please comment more on the reasons why cooling may be higher or lower depending on the case and/or location?

Carrió, D. S., Homar, V., Jansà, A., Romero, R. and Picornell, M. A.: Tropicalization process of the 7 November 2014 Mediterranean cyclone: numerical sensitivity study, Atmospheric Research, 197, 300–312, 2017.

Davis, C. A., & Bosart, L. F. (2004). The TT problem: Forecasting the tropical transition of cyclones. Bulletin of the American Meteorological Society, 85(11), 1657-1662.

Di Muzio, Enrico, Michael Riemer, Andreas H. Fink, and Michael Maier‐Gerber. "Assessing the predictability of Medicanes in ECMWF ensemble forecasts using an object‐based approach." Quarterly Journal of the Royal Meteorological Society 145, no. 720 (2019): 1202-1217.

Fita, L., & Flaounas, E. (2018). Medicanes as subtropical cyclones: the December 2005 case from the perspective of surface pressure tendency diagnostics and atmospheric water budget. Quarterly Journal of the Royal Meteorological Society, 144(713), 1028-1044.

Flaounas, Emmanouil, Fanni Dora Kelemen, Heini Wernli, Miguel Angel Gaertner, Marco Reale, Emilia Sanchez-Gomez, Piero Lionello et al. "Assessment of an ensemble of ocean–atmosphere coupled and uncoupled regional climate models to repro-duce the climatology of Mediterranean cyclones." Climate dynamics 51, no. 3 (2018): 1023-1040.

Miglietta, M. M., Cerrai, D., Laviola, S., Cattani, E., and Levizzani, V. ( 2017), Potential vorticity patterns in Mediterranean "hurricanes", Geophys. Res. Lett., 44, 2537– 2545, doi:10.1002/2017GL072670.

---

## Author Comment (AC1) · 19 Feb 2020

Response to Reviewers Comments, Surface processes in the 7 November 2014 medicane from air–sea coupled high-resolution numerical modelling, Bouin and Lebeaupin Brossier, Atmos. Chem. Phys. Discuss.

We thank both reviewers for their detailed comments which helped us to substancially revise this paper. Here below is a summary of the main changes we made.
- the introduction was lengthy, not very clear due to technical wording and not focused on the subject of the paper, i.e. the impact of coupling with the ocean and the role of the surface fluxes. We removed some unnecessary details and added some information on the lack of clear medicane definition.
- the time evolution of the cyclone and its different phases are assessed in a more objective way, based on the low-level temperature field and upper-level PV anomaly. This leads to a slightly different timing of the event, but does not change the conclusions. Two figures seemed unnecessary and were removed, a figure showing the overall time evolution has been added, a figure comparing the rainfall rates during two phases has been added.
- the description of the different phases, in term of surface versus upper-level mechanisms has been shortened and merged with the analysis of the surface fluxes (Section 4). Only a short description of the cyclone chronology has been kept in Section 3.
- we provide more details on how our results (in terms of cyclone track or thermal winds) compare with other results on the same case.
We provide below detailed answers to the reviewers comments.

AR #1

Major points:
- The distinction among the three different phases in the cyclone lifetime is subjectively defined, thus not completely convincing. I recommend you find an objective way, e.g. based on the Hart (2003) diagram, or on the methodology discussed in Fita and Flaounas (2018).

Yes, this is an important comment, and we realized that the classification was probably not appropriate. The phase diagram, in this case, was not sufficient to distinguish between the different phases. We choose to use the upper level and low level dynamics (300 and 850 hPa) as in Fita and Flaounas (2018) (their Fig. 6) to investigate the cyclone evolution and separate the different phases in a more objective way. The description of the chronology of the medicane (section 3.1) is now based on a figure similar to the one of Fita and Flaounas (2018) and on the Hart (2003) diagram. The latter has been changed to make it similar to the one of Fita and Flaounas (2018) and to make the different phases easier to distinguish. The evolution of the medicane follows 3 phases, development, mature phase and decay, they are time shifted with respect to the previous classification. One figure has been added (Fig. 5), all the figures showing time series has been redone with colour codes corresponding to this new chronology, and the discussion about the evolution of the air-sea heat fluxes and the respective roles of the surface parameters has also been changed. The new time periods we consider (in Tables 1 to 3 and Figures 13, 16 and 18) are representative of these phases of the event. We feel that this chronology based on more objective criteria is more robust and that the corresponding analysis is easier to follow.

- The discussion is Section 3.1 is not convincing in many points (see also minor points in 3.1.1 and 3.1.2):
L280-285: the bands of colder air may not be due to the evaporation of precipitation. In particular, the one to the left may be due to cold-air advection, which secludes the cyclone warm core, turning around the center within the cyclonic circulation, as suggested by Fig. 6a. To clarify this aspect, one should perform numerical simulations without the evaporation of precipitation: this would

demonstrate whether the cold air is due to a long-range transport toward the center of the cyclone or really to the evaporation of precipitation.

We understand your point, so numerical simulations without the cooling due to the evaporation of the precipitation were done, and the comparison of the virtual potential temperature at low level shows clearly that the bands of colder air are actually due to the rain evaporation (see Fig. R1 without cooling with respect to Fig. R2 with cooling).
We feel not necessary to add this figure in the manuscript, but we now mention the results of this additional run as an evidence of the origin of these cold pools, versus the cold air advected from the North African coasts, which are still present in the simulation without the cooling due to rain evaporation.

[Figure]

Figure R1:  Map of equivalent potential temperature (warm colors) and virtual potential temperature below 19 °C (blue shades), and SLP (black contours) at 08:30 UTC on 7 November, when the cooling due to rain evaporation is turned off.

[Figure]

Figure R2:  Map of equivalent potential temperature (warm colors) and virtual potential temperature below 19 °C (blue shades), and SLP (black contours) at 08:30 UTC on 7 November.

L286-288: I understand that your focus is on surface fields, but you should also consider what happens in the levels immediately above (e.g., 950 hPa, 900 hPa) to support your considerations and better identify the origin of the different air masses and thermal gradients. The vertical extension of the cold air masses in North African may be limited to a few meters, so you should demonstrate more clearly that the "advection of cold and dry air . . . from the Tunisian and Libyan continental surface . . . (L296-297, L448-449)" is relevant. For example, you can use backtrajectories, going earlier in time than those shown in Fig. 9, to clearly illustrate the origin of the air parcels.

We are not sure to understand this comment, as the vertical extension of the air masses both in the cold pools due to rain evaporation and those advected from North Africa is visible in the vertical cross section in (former) Fig. 7b. This figure has been reduced in the new version, as we think that two vertical cross sections were redundant to illustrate the role of cold pools. We keep the E-W section, that clearly shows, in consistency with (former) Fig. 7a, the extension of both the cold pools due to evaporation (between 500 and 1000 m) and of the cold air advected from North Africa (between 1000 and 1500 m).

Minor points:
Lines 9-12: "The deepening . . . of the cyclone is due . . . then to low-level convergence and uplift of conditionally unstable air masses by cold pools, resulting either from rain evaporation or from advection of continental air masses from North Africa.": the deepening can be due to WISHE mechanism and/or to baroclinic processes, the way precipitation is generated is related to convergence and uplift by cold pools, so the sentence should be reformulated.

Yes, this has been reformulated, "Heavy precipitation result from uplift of conditionally unstable air masses due to low-level convergence at sea"

Line 14: . . . due to a sea surface cooling . . .

Yes, change done.

Line 27: at least up to the mid troposphere . . .: really, even in the version of Hart diagram modified by Picornell et al. (2014) you used, the vertical structure is analyzed also in the upper troposphere, up to 400 hPa.

Yes, we analysed the vertical structure up to 400 hPa, but there was a typo in the legend of Fig. 4. This has been corrected.

Line 33: note that the 26 ◦ C threshold of tropical cyclones cannot be applied to tropical cyclones developing from tropical transition processes (see McTaggart-Cowen et al., 2015).

Yes, we added the reference, and a short comment on that. Thank you.

Line 41, 274: I image you are referring to lee cyclones (see Tibaldi et al., 1990), not to lee waves. Lee cyclones are not only the effect of a wave, but theory provides a comprehensive way to describe the process of cyclogenesis.

This has been changed to "lee cyclones forming south of the Alps or north of the North African reliefs (Tibaldi et al., 1990)", thank you for the reference.

Line 119: . . . its initiation as a baroclinic storm . . .

This sentence has been removed.

Section 2.1: I understand that 19 Figures are a lot, but please consider the possibility to include additional 1-2 figures to make easier to understand the results in this Section, or refer to Figures published in previous papers.

Yes, we realized that this part is not easy to follow. We added one figure (Fig. 1) showing the upper-level PV anomaly, SLP and temperature at 850 hPa from the ERA-5 outputs at two epochs to make the large-scale description of the chronology of the event easier to follow.

Line 144: please refer to Fig. 1 to help following the text.

Yes, we refer here to (now) Fig. 3.

Line 170: Is convection treated explicitly? Please add this piece of information.

Done "In this configuration, deep convection is explicitly represented while shallow convection is parametrized using the eddy diffusivity Kain-Fritsch scheme (Pergaud et al., 2009)."

Line 171: grid spacing instead of resolution.

Change done.

Line 189: what is an ORCA grid?

We added the piece of information "(tripolar grid with variable resolution, Madec and Imbart, 1996)".

Section 2.2.3: what is the rationale behind the choice of this domain extension? Have you tried with different domains? Apparently, only a small domain extension is favorable for a proper simulation of the track - cfr. Cioni et al. (2018) and your tracks with those in Carriò et al. (2017) and Pytharoulis et al. (2018). Please, comment on this.

The only test we made was to use only one domain with a resolution of 2 km, boundary forcing and initial conditions from the ECWMF analyses, without grid nesting. The domain extension, however, was similar to the D2 extension used in this study. We obtained a better track, but no deepening of the cyclone. The present D2 domain has been chosen to cover the cyclone evolution, including the surrounding coasts. Possibly, a simulation with a smaller nested domain with finer resolution like in Cioni et al. (2018) would lead to a better simulated track, but we did not test this (also because we wanted our resolution to be comparable with the one of the operational AROME forecasts, 1.3 km). We added a comment on this "This domain extension was chosen as a trade-off between computing time and an extension large enough to represent the physical processes involved in the cyclone lifecycle, including the influence of the coasts".

Figure 3a: it would be helpful to add at least the maximum wind recorded in Lampedusa, Malta, Pantelleria in panel a.

Yes, they were added.

L274: . . . by the lee cyclone induced by the North African relief . . .

Yes, change done.

L279-280: note that it is not the convergence between SE flow and S-SW flow responsible for the most intense precipitation at sea in Fig. 6b, but the one associated with the low-level southwesterlies and the northwesterly flow behind.

We were referring here to the convergence line east of the domain, at the cold front.

L284: northwestwards?

We are referring to the cold air advected from the African coasts, and the flow is actually northeastwards.

L285: deep convection in . . .

This paragraph has been removed.

L292: The low-level virtual . . .

Yes, change done.

L300: in the southeasterly low-level flow . . . (in Fig. 7b, the wind component seems from the south).

Part of the flow (in the SE of the cyclonic centre) is southeasterly and part of it (NW of the cyclonic centre) is northwesterly, that is why low-level convergence at sea is strong. We were referring here to the NW flow.

L302-305: did you check these points in the simulation? May you be more quantitative?

The dry air intrusion with the values of the RH were checked, they are visible Fig. 15, and we now refer to it. The other points were not, and we preferred to remove them.

L306: the high CAPE is not obtained by extracting heat and moisture! The surface fluxes modify the low-level features, determining an environment more favorable to instability (i.e., CAPE increased).

Yes. This paragraph has been removed, due to a major comment of Reviewer #2.

L316-318: again, this sentence should be less qualitative in order to be more convincing.

Actually, this sentence has been removed, as we feel it unnecessary after the more quantitave description based on Fig. 15. The role of latent heat release has been assessed using the backtrajectories and the separate evolution of potential temperature and humidity along the trajectories and this is detailed in sect. 4.4.

L323: Mazza et al. (2017) does not refer to the December 2005 Medicane.

Yes, thank you for checking, this has been corrected.

L331: you really show the role of air-sea interaction processes; the role of diabatic processes may be inferred from theory, you do not show that it occurs in this case.

This part has been changed, we added an assessment of the temperature and humidity evolution during the particle travel at sea first, then during its uplift. This shows more clearly that the latent heating is at play during the uplift of the particle.

L342-343: I suggest to change "the evolution of the SST" with "sea currents".

Really, the object of section 3.2.1 (now 3.2) was to check the evolution of the SST in the coupled simulation and its impact on the atmosphere. The currents play a secondary role. As this part has been changed, the sentence now reads "In the following, the possible impact of the ocean−atmosphere coupling on the cyclone intensity is examined by comparing the results of the CPL, NOCUR, and NOCPL simulations. The time period for this comparison is the 7 November only, as the medicane has lost a large part of its intensity in the evening of the 7 November."

L360: an important conclusion you could mention here is the negligible role of currents.

The negligible impact of the currents was mentioned l. 277, l. 294, l. 310. We add a sentence at the beginning of the discussion "Coupling with the surface currents has no significant impact of the simulation". We also add a short sentence "Surface currents have no impact" in the abstract.

L384: the gradient of wind speed . . . between which levels?

We changed the sentence to " the wind speed at first level with respect to the sea surface".

L385: I suggest to reformulate as "humidity at saturation with temperature equal to SST".

Yes, done.

L394-398: this part is already discussed and can be omitted.

Yes, removed.

L410-411, L421: This is partly due to the conditional sampling . . .: explain better.

What was meant here is that the lower tail is cut by the high-pass filtering. We added " as low fluxes are cut off by the sampling" and hope this is clearer.

L424-425, L439: The sentences "The time evolution of the distributions of LE and the SST are opposite to each other" and "parameters controlling LE (and evaporation) are the SST . . . with very strong positive correlations" do not contradict somehow each other?

Yes. The first sentence was misleading, and is a (classical) confusion between apparent correlation and causality, so it has been removed. Thank you for that!

L442: please change "globally" into "positive".

This paragraph has been removed, due to a major comment of Reviewer #2.

L442-443: this sentence is not clear; my interpretation is that you should rearrange in something like: "The fact that r is low in the whole domain, and higher in EF600 suggests that strong evaporation controls specific humidity and temperature".

This part has been re arranged due to comments of Reviewer #2, and this sentence has been removed.

L448-449: see major point.

We hope that the reply to this major point is convincing.

L485: in contrast . . .

Change done

L485-486: again, see major point.

We hope that the reply to this major point is convincing.

L531: . . .lack diabatism . . .: really, the second case discussed in Miglietta and Rotunno (2019) does not lack diabatism, but contains both diabatism and baroclinic processes. Please clarify this point here and later in the discussion (L561-564 should be changed, as the Medicane apparently belongs to the second category of Medicanes).

The sentence has been changed and now reads: "Medicanes of the second category also present similarities with tropical cyclones, like deep warm core and symmetrical wind field, but present both diabatism and baroclinic processes". We also make clear later on in the discussion that "This suggests that the medicane of November 2014 as simulated in this study presents characteristics close to an extratropical cyclone, or medicane of the second category as in Miglietta and Rotunno (2019).".

L555: This is consistent with the observations in Miglietta et al. (2013) and Dafis et al. (2018).

We added a sentence at the end of this paragraph "The convective activity is stronger during the development than during the mature phase of the cyclone, resulting in heavy rainfall 12 to 6h before the maximum wind speed, in consistency with previous studies of medicanes based on observations (Miglietta et al., 2013; Dafis et al., 2018).". Also, a figure has been added (Fig. 7) to compare the mean rainfall rates between the development and mature phases.

Figure 6 is very confusing: the coastline can be hardly identified; latitude and longitude are not reported; the extension of the cross section in b) is not clear; finally, the same color scale should be used in both panels. Similar considerations apply to Figure 7.

Yes, we realized that these figures are not easy to read. Fig. 6 has been removed, as we felt it not necessary to the argumentation, and Fig. 7 (now Fig. 11) has been simplified to keep only the necessary information: equivalent and potential temperatures, wind, SLP and strong convergence. Also, the coastlines have been made easier to see and the latitudes/longitudes have been added.

Figure 9: how long do the backtrajectories go back in time?

They start at the beginning of the D2 simulation, at 00 UTC on November, and this is now specified in the caption.

Figure 10: the cyan and blue columns are difficult to distinguish.

We increased the shift between those and we hope this is easier to read.

Figure 11: the triangles are difficult to identify.

We increased the symbol size.

AR #2 (E. Flaounas)

Major comments 1) I find that the technical language used to describe atmospheric dynamics in several parts of the text comes at the cost of a clear interpretation of cyclone dynamics. Here are several examples from the abstract and introduction. I suggest a more careful revision of technical language throughout the text.

Yes, we realized that there was plenty of unnecessary technical vocabulary, especially in the Introduction and Section 3 describing the chronology of the event, and that it does not help to describe the cyclone dynamics. We tried to rephrase this. Thank you for your careful rereading.

line 10: "Tropicalization". I am aware of "tropical transition" but I am afraid I am not familiar with this term. What is its physical/dynamical content? If this term has been used before, then its physical content could be very briefly explained alongside with a reference, if not then, maybe it should be defined in the text.

Tropicalization was used in the paper of Carrio et al., 2017: "Tropicalization process of the 7 November 2014 Mediterranean cyclone: numerical sensitivity study". It is maybe not very common, so it has been replaced by tropical transition as suggested.

line 10: ".. then to low-level convergence and uplift of conditionally unstable air masses by cold pools..". Is it meant that "tropicalization" is a product of jet crossing and latent heat release? Then from the perspective of cyclone dynamics maybe it should be mentioned that both baroclinic and diabatic forcings act in synergy to deepen the cyclone.

Done, thank you for suggesting that, it now reads: " The deepening and tropical transition of the cyclone results from a synergy of baroclinic and diabatic processes".

line 12: "...feeding the latent heat release during the mature phase of the medicane..." do you mean that specific conditions favour convection, in turn leading to high latent heat release? Such phrases were found in several other parts of the manuscript.

This has been rephrased to:" Backtrajectories show that air-sea heat exchanges moisten the low-level inflow towards the cyclone centre".

line 40: A PV streamer is filament of high PV flow it may not be always regarded as equivalent to a trough.

We made it clearer, using " an elongated upper-level trough".

line 40: "potential vorticity (PV) from higher-latitude regions.." you mean intruding stratospheric air characterized by high PV values?

This has been rephrased to: "cold air with high values of potential vorticity "

lines 40-43: I am not sure how these lines help us understand the contribution of different processes in medicanes development.

We wanted to introduce the discussion on the processes due to the specificities of the Mediterranean in the medicane lifecycle, so we listed several geographic factors contributing to their development. The sentence has been made clearer: "From these studies, a feature common to many medicanes is the presence of an elongated upper-level trough (also know as a PV streamer) bringing cold air with

high values of potential vorticity (PV) from higher-latitude regions. Other local effects favouring their development are:  lee cyclones forming south of the Alps or north of the North African reliefs (Tibaldi et al., 1990); impact of the coastal reliefs in triggering deep convection; and relatively warm sea surface waters able to feed the process of latent heat release during their mature phase.”

line 46: what do you refer by "upper-level thermodynamics". What is meant by "limited", maybe you mean "inadequate"?

Yes, this sentence was not clear. We rephrased to “It is nevertheless inadequate to describe the respective roles of upper-level and low-level processes (e.g.  surface heat exchanges  or role of geographical conditions like orographic lifting).”

Line 49: "behaviour", "extension", "..various characteristics" please be specific by rephrasing or attributing a physical content to each term. Please also note that Fita and Flaounas (2018) that warm seclusion may be diagnosed as a warm core symmetric system even if convection is weak. This is opposed to tropical cyclone dynamics.

Yes, this has been changed to “a large diversity of duration, extension (size and vertical extent) and characteristics (dominating role of baroclinic versus diabatic processes)”. We are aware that warm cores may exist as a result of warm air seclusion, this is why we used backtrajectories to check that air is warmed first by air-sea enthalpy fluxes then during convection (diabatic process).

Line 53: "PV transfer from above" I am not sure what is meant here. Do you suggest that a cyclone was intensified due to the formation of a PV tower after aligning air of stratospheric origin with diabatically produced PV in the lower to mid troposphere?

Yes, but the formulation was not clear. It now reads: “ rapid deepening of the surface low-pressure system by interaction between low troposphere and upper-level PV anomalies”

Line 56: As formulated here it seems that the PV streamer and the upper level Jet are two different atmospheric features.

Yes, we agree that the formulation was misleading. We reformulated to “ Recently, the ubiquitous presence of PV streamers and their key role..”.

Line 60: "large diversity of characteristics and behaviours". Please provide physical content to these words.

This sentence and the following one have been replaced by “They concluded also that, during their lifecycle, medicanes can rely either on purely diabatic processes or on a combination of baroclinic and diabatic processes.”

Line 61-62: I am not sure I understand this phrase. In main lines PV anomalies in a given location can originate from advection (adiabatic process) or from momentum/heat exchanges with the environment (diabatic). Both contribute to the formation of a PV tower and thus both provide a combined baroclinic or diabatic forcing to cyclonic circulation.

Please see above.

Lines 74-77: This phrase is long and difficult to understand.

This has been rephrased: "Turning off the surface turbulent fluxes during different phases of the cyclone brought contrast to this view, showing that the role of surface enthalpy in feeding the cyclonic circulation is not constant throughout its lifecycle. Indeed, it revealed important during its earliest and mature phases, playing only a marginal role during the deepening (Moscatello et al., 2008, case study of September 2006)."

Line 76: please note that atmospheric processes (e.g. convection, or baroclinic instability) are expected to form, or sustain a vortex/cyclonic circulation not enthalpy or surface thermodynamical variables. What is meant here is that surface processes enhance convection and thus a diabatic forcing to cyclonic circulation?

Yes, we made it clearer with " the role of surface enthalpy in feeding the cyclonic circulation".

Lines 82-83: "WISHE-type vs baroclinicity". Please rephrase. Baroclinicity is not a mechanism is an atmospheric state.

Yes, we rephrased to "baroclinic processes".

Lines 83-84: It is quite awkward to characterize a PV-streamer as "instability aloft". Do you mean that stability decreases within the atmospheric column under the PV-streamer?

We rephrased to " high upper-level PV values".

Lines 85-86: I am not sure I follow the rational here. Fita and Flaounas, (2018) present a detailed analysis of the implicated dynamics in the development of the december 2005 case. Both a PV-streamer and a cut-off low were involved (the former evolved to the latter). Both features correspond to a PV anomaly of statospheric origin (cut-off or streamer) that provide a baroclinic forcing to the cyclone. I would say that the difference comes from the extent that baroclinic or diabatic forcing (basically latent heat release due to convection) prevails in provoking cyclonic circulation.

What was meant here is that in the first case, the upper-level PV anomaly above the cyclone centre was connected to the large-scale environment, while it was isolated as a cut-off low in the second case. We agree that a medicane can present both features at different stages of its lifecycle (as this is the case in the present study), and we removed this part of the sentence, as it was misleading and not necessary.

Lines 82 & 87: "WISHE-type" and "WISHE-like". Please be consistent on the terminology. If the mechanism explained here diverges from the original WISHE, please be more specific.

The terminology has been homogeneized, thank you for checking.

Lines 87-88: I do not understand how warm seclusion "replaces" the "heat and moisture to latent heating through a WISHE-like mechanism". Is it meant here that the low-level warm core in the first case is due to convection and latent heat release, while in the second case it is due to warm seclusion?

Yes, this is what was meant. We rephrased to "In the case of October 1996, the cyclone warm core is formed by latent heat release fed at low level by heat and moisture extracted from the sea. In the December 2005 case, the warm core is due to warm air seclusion."

Line 90: Would you agree that the "Jet-crossing" mechanism refers to a baroclinic forcing to the cyclone? E.g. Jet-crossing could deepen a cyclone even in the absence of any diabatic processes. This seems to be similar to the upper tropospheric forcing "DΦ" on the medicane case of 2005 in Fita and Flaounas (2018), their Figure 7.

Yes, we agree with that. The sentence has been removed, as confusing and not necessary.

Line 98: Intensity may change due to processes such as convection, not surface heat fluxes. A warmer SST would suggest enhanced convection or at least, favour latent heat release.

Yes of course, but enhanced convection may result from higher CAPE values at low level, due for instance to stronger enthalpy fluxes at the surface. This was simplified to " warmer (respectively colder) SSTs lead to more (resp. less) intense cyclones"

Line 108: Please also note that the impact of coupling has limited effect on "all intense Mediterranean cyclones". This was shown in a simulation ensemble by Flaounas et al., (2018). This could reinforce the argument that known medicane cases are not expected to have an exceptional sensitivity to air-sea coupling than the rest of the intense cyclones.

Yes. The results of Flaounas et al. (2018) are consistent with those of Gaertner et al. (2017), but the latter authors mentioned that this lack of impact can be due to the relatively low resolution of the simulations they compared. If we understood correctly, the simulations used in Flaounas et al. (2018) have also resolutions between 18 and 50 km. We added the reference.

2) Medicanes definition and relevance with this case: A major drawback in the field is the lack of a physical definition for medicanes. I am not sure that it has ever been shown that there is a "perfect vertical alignment between the sea level pressure (SLP) minimum". I could argue that in several papers there is no such alignment observed, especially when the "eye" is fully developed.

Yes, vertical alignment is more a sufficient than necessary condition, and this part of the sentence has been removed.

In line 30 how is radius defined?

We referred here to radius of maximum wind as for the tropical cyclones. We realized that there is probably not appropriate, several authors using the radius of warm core anomaly at 600 hPa for instance. We changed this to "radius" and added the citation of Picornell et al. (2014) as a reference on the typical size of medicanes.

I suggest a more careful opening paragraph mentioning that knowledge comes for a very limited number of cases qualified as medicanes where no objective criteria have been used other than some arbitrary visual characteristics of cloud coverage. After all, several papers show that medicanes present a high variability of dynamical structures (e.g. Miglietta et al., 2017).

Yes, a paragraph on that has been added in the beginning of the introduction: "The medicane cases confirmed by converging characteristics as those mentioned above represent only a small portion of the Mediterranean cyclones (e.g. 13 over 200 cases of intense cyclones in the study of Flaounas et al., 2015, or roughly one per year). Due to this scarcity, isolating in a definite way the characteristics enabling to separate medicanes from other Mediterranean cyclones proved elusive. A study using dynamical criteria concluded that medicanes are very similar to other intense cyclones, with a slightly weaker upper-level and a stronger low-level PV anomalies (Flaounas et al., 2015). Recent comparative studies (e.g. Akhtar et al., 2014; Miglietta et al., 2017) showed a large diversity

of duration, extension (size and vertical extent) and characteristics (dominating role of baroclinic versus diabatic processes) within the medicane category ".

In fact, from a climatological context, medicanes are not expected to clearly stand out in terms of dynamics from several other cases that were not "qualified" as medicanes.
This seems to be one of the results for several of the metrics used by Tous ad Romero (2013) and Flaounas et al., (2015) in line 37.

Yes, we hope that the previous paragraph makes that clear.

3) I believe that cyclone dynamics in section 3 are not convincingly addressed (this also concerns section 5). It seems to me that this is also partly due to how technical language is used (comment #1).
In summary a PV streamer intrudes the Mediterranean, it provokes cyclogenesis and wraps around the cyclone centre until it evolves to a cut-off at 300 hPa (this is not very clear due to the domain size in figure 5). This scenario seems to be fairly similar to the 2005 medicane, analysed by Fita and Flaounas (2018) and Miglietta and Rotunno (2019), but also to several other cases.

In the "transitional" phase (Fig. 5b), the main body of the PV anomaly is dislocated with respect to the cyclone centre and overall I would say that this example complies with the classic paradigm of baroclinic instability in Hoskins et al., (1985), as illustrated from a PV perspective. What do you refer to with "no trace of baroclinicity" in line 265? With such high PV anomalies in the upper troposphere, the potential temperature isosurfaces should slope accordingly, creating a barcolinic environment to the proximity of the cyclone.

We meant that the frontal structures at low level were not detectable anymore, but we agree that this is a shortcut. We rephrased the sentence to "The medicane presents the circular shape typical of tropical cyclones with spiral rainbands, and a warm, symmetric core (Fig. 5d) extended up to 400 hPa (Fig. 6)."

The analysis in 3.1 is based on the PV-streamer evolution, which mainly refers to adiabatically advected stratospheric air masses. Why do you refer to the last phase of the section as "diabatic"?

How can rainfall be indicative of diabatic vs baroclinic forcing to the cyclone intensification?

We meant here that the heavy precipitation are due to deep convection, which is co-localized with strong PV anomalies at low level and should correspond to with latent heat release (diabatic process), but we agree that this is a shortcut, and contradictory with the heaviest precipitation occuring during before this phase. The description of the chronology of the medicane has been shortened and this sentence has been removed.

Actually the presence of high PV anomaly in the upper troposphere suggests a considerable forcing to the surface cyclone.

Yes, we were referring to the low-level processes here.

Figures 6 and 7 are overcharged and not consistent in the plotted fields. It is also quite odd to use potential temperature at surface level. If this is a model diagnostic (e.g. as 2-meter temperature) rather than a prognostic variable, then it is not used by the model to resolve dynamics. This would create some inconveniences in the analysis.

Furthermore, the field is strongly influenced by land-sea transitions. I suggest you use 850 hPa for low level baroclinic structures. Finally, the two overlaid temperature fields in Fig. 7 makes interpretation of dynamics rather difficult.

Figure 6 appears not necessary and has been removed. We kept Fig. 7 (now 11) to show the role of the cold pools created from rain evaporation or advected from North Africa is initiating the convection. Is has been simplified to show only the necessary fields, namely the equivalent and potential temperatures (we felt that these two fields superimposed are actually useful to demonstrate the effect of the cold pools, see of the major comments of Reviewer #1), the SLP, surface wind and strong low-level convergence. It also shows the vertical extension of the cold pools. The potential temperature is a prognostic variable of the model, but equivalent potential temperature and virtual potential temperature are diagnostic. We chose to look at the temperatures at the first level of the model to represent the horizontal extension of the cold pools.

Lines 274-275 "baroclinic processes are responsible for the heavy precipitation" I am not sure I understand this phrase. Are you suggesting that large scale, quasi-geostorphic forced ascent provokes rainfall?

We meant that the frontal structures present at the beginning of the simulated medicane result in convergence and ascent of warm and moist air masses, then in precipitation. This can be interpreted as large-scale rather than more local forcing, but also as baroclinic (frontal) rather than barotropic (e.g. orographic) forcing.
The formulation is now "The heavy precipitation obtained during the first hours are co-localized with frontal structures."

Section 3.1.3: I do not really understand the term "diabatism". If you refer to the mechanism of tropical transition (Davis and Bosart, 2004) then I guess that you suggest that an initially baroclinic system is now maintained due to convection (as stated in line 316). However there is still an upper PV anomaly due to the streamer. How much does it contribute to the surface cyclonic circulation (also evident in the PV profiles of Fig. 18)?

Yes, the upper level PV anomaly is present throughout the whole event, and it likely contributes to the cyclonic circulation down to the surface, but this has not been quantified, as the main subject of the study is the role of the coupling with the sea surface and the role of the different surface parameters in controlling the enthalpy fluxes.

Does the absence of divergence in the upper troposphere suggest that the dynamics are different from the ones expected by tropical cyclones (lines 313)?

Yes.

It seems that the trajectories "confirm that the warm core of the medicane is actually due to diabatic processes. (lines 330-331)". How do high θe values (regardless the origin of these values) assure that the cyclone is maintained due to diabatic forcing?
Equivalent potential temperature (θe) should remain constant if condensation takes place. Here, air masses experience a dramatic increase of θe. Does this reflect an increase in heat fluxes or moisture?

High θe values do not ensure that the cyclone is maintained due to diabatic forcing. Here we follow some particles experiencing heating (increase of θe) first at low-level when in areas of strong surface enthalpy fluxes (south and east of Sicily), then when uplifted by deep convection. We did not demonstrate quantitatively that this is due to diabatic heating, and the value of this heating. We

corrected this by assessing the respective changes in humidity (mixing ratio) and potential temperature along the trajectories. During the "surface enthalpy flux stage" i.e. when the particles are close to the surface (between 250 m and 1000 m above sea level), the potential temperatures decrease (-1 °C in average for the particles show Fig. 17) while the mixing ratio increases of 4.5 g kg$^{-1}$ in one case, 2 g kg$^{-1}$ in the 2 other cases. This shows that the change of $\theta e$ is due to latent heat flux (evaporation) at the surface. During the convective stage, when the particles are uplifted from a few hundreds meters to 1500 m, their mixed ratio decreases of 2 g kg$^{-1}$ and their potential temperature increase of 3.5 to 5.4 °C (condensation and heating, which can reasonably be interpreted as latent heat release). We added a paragraph on that in the text, with the figures mentioned above.

Furthermore, how often do the air masses hit the ground and what is the time interval used to calculate the trajectories? Could you comment on how the method used to calculate the trajectories assures that the "same" air mass is followed in time (especially given the sudden change of altitude after leaving the Sicilian coast)?

Particles followed by the backtrajectories module, in this case, do not hit the ground, they travel between 200 m and a few km. The method used here is based on a passive tracers method (Gheusi and Stein, 2005) inspired by the work of Schär and Wernli (1993). The reference has been added in the text.
Backtrajectories enable to follow the same particle from the start of the simulation to this end. It does not ensure, of course, that this particle does not experience transformation, like a change of its potential temperature, humidity, or state.

In fact, factor separation technique in Carrio et al., (2017) shows that the development of this cyclone is due to synergy between upper tropospheric forcing and latent heat release (their figure 14). Could you compare and discuss with respect to their results?

Indeed, the study Carrio et al. (2017) concludes that the upper level PV anomaly is a key process in the preconditioning phase, and that the synergy between this and the latent heat release leads to the rapid deepening of the event. Our conclusions are close to those, and we added a paragraph on that in the discussion: "Finally, these results are consistent with those of Carrio et al. (2017). By using a factor separation technique, they show that while the role of the upper-level PV anomaly is crucial in preconditioning the event, its rapid deepening is due to the synergy of latent heat release and upper-level dynamics".

4) I suggest to reorganise the manuscript structure to make it more attractive to the reader. Section 4 is very intensive, the importance of the results is eclipsed by a plethora of diagnostics and variables and seems quite detached from section 3. I would suggest to the authors to make a clearer and more refined presentation of the objectives in the introduction and merge the sections 3 and 4 by organising the document sections according to cyclone phases. In fact, section 5 seems to include quite interesting analysis that would be more adequate for section 3.

Yes, this proved to be a very useful comment, helping us to clarify the presentation of the results and to discriminate between our main and secondary results. We re organize the manuscript with only a short description of the chronology of the event in Section 3, more objectively derived from the upper-level and low-level dynamics as in Fita and Flaounas, and completed by the Hart (2003) diagram. We keep in this section the analysis of the SST evolution and of its negligible impact on the medicane. We re arrange Section 4 to include both the results on the control of the surface heat fluxes by the parameters and a more detailed description (than in Section 3) of the different phases of the event. This description is both shorter and more quantitative than is the previous version of the manuscript. Especially, we keep mainly what is related to the low-level processes, and necessary

to understand the progress of the medicane. We use the temperature and humidity evolution in a separate way along the particles trajectories (see above) to check that latent heat release is actually one of the processes of this medicane.

Minor comments:
Abstract: One important message is that intense air-sea interactions have not a strong effect on the cyclone development. Please further highlight your main results, I am not sure about what message we get from lines 16-21.

This has been rephrased as "Analysing the surface enthalpy fluxes shows that evaporation is controlled mainly by the sea surface temperature and wind. Humidity and temperature at first level play a role during the development phase only. In contrast, the sensible heat transfer depends mainly on the temperature at first level throughout the medicane lifetime."

Line 131: It seems from Di Muzio et al., (2019) that the occurrence of Qendresa was well predicted by ECMWF with a lead time of 7-8 days in contrast to most medicanes, where predictability seems to be possible only in erlier lead times. Could you please comment, or be more specific, on the predictability of this case?

Indeed, the work of Di Muzio et al. (2019) concluded that Qendresa showed a good predictability as early as 7.5 days lead time (LT). This predictability is defined in their paper as the probability of occurrence of the cyclone, based on the ensemble forecasts, with respect to the operational analysis results. They also assessed the predictability of other parameters, like the position, the central pressure, the symmetry, compactness and upper-level thermal wind, mentioned as "the most relevant parameter in distinguishing TLC from fully baroclinic cyclones". In the case of Qendresa, they obtain a "position forecast jump" (increase of the number of members correctly representing the position of the cyclone during its intense phase) at 4 days LT, but their median position error increases when the LT decreases, to reach a relative maximum at 1 day LT (their Fig. 6), close to the starting time of our D1 simulation. This is consistent with the poor quality of the simulated track we obtained both in the D1 and D2 simulations. Also, the central pressure they obtain (Fig. 4) is ~ 10 hPa higher in average than the analysed one, and they note that "storm intensity forecasts [are] showing little convergence towards the analysis value for Qendresa". We added the following paragraph in the text: "A recent study based on the ensemble forecasts of the ECMWF (Di Muzio et al., 2019) showed that the predictability of occurrence (with respect to the operational analysis) is good as early as 7.5 days lead time, but the predictability of the position in weak, especially between 4 and 1 days lead time (their Fig. 6). The predicted central pressure is also consistenly 10 to 14 hPa higher than the analysed one, whatever the lead time considered".

Line 213: Typically trajectories refer to air masses and tracks to cyclones. What "best-track" refers to and what kind of observations were used?

The best track, in the present case as in the study of Carrio et al. (2017) was derived from the SEVIRI 10.8 m channel brightness temperature, this is now specified in the text. Also, the confusion between track and trajectory has been corrected.

Line 226-227: This is contradictory statement please rephrase. In fact, the northward displacement of the track only allows to confirm an agreement of the deepening phase of the cyclone between observations and model.

This has been changed to "The simulated cyclone nevertheless shows a deepening and maximum intensity close to the observed ones".

Line 239: I do not follow the reasoning of using 100 km radius, been "fitted to the radius of maximum wind". The 100 km radius used is too small, compared to the size of the cyclonic circulation. This is quite clear from the wind field in e.g. Fig. 6. Therefore it seems quite difficult for the diagnostic to capture asymmetries due to frontal structures, which in turn develop in higher length scales. Could you please comment on this?

The cyclone size (defined as the radius of maximum wind at 850 hPa) strongly decreases between the beginning of the development phase and its end (as soon as the cyclone gains it circular shape). To be consistent with the mature and decay phases, we chose the 100 km radius, close to the radius of maximum wind (90 km). As a result, the structure of the cyclone is not well represented by the Hart (2003) diagram during the first part of the development phase, this is why we use the upper-level and low-level dynamics as in Fita and Flaounas (2018) to characterize the time evolution of the cyclone, rather than the Hart (2003) diagram. We added a comment on the size issue in the text: "Please note that the radius of maximum wind is ill defined or larger during the first stage of development of the cyclone, whereas it is steady and close to 90 km during the major part of its lifetime. As a result, the diagram obtained is probably not representative of the cyclone structure during its first hours."

Furthermore Di Muzio et al., 2019 used the operational analyses and same set-up for cyclone phase diagrams and argues that Qendresa presented no warm-core in the upper troposphere. Could you comment on the sensitivity of phase diagrams in your case?

We suggest that the phase diagram representation is sensitive to the model resolution. In the case of the operational analyses used in the work of Di Muzio et al. (2019), the horizontal resolution is 16 km and, given the small size of this event, this could explain the low value (-14) of the upper-level thermal wind in the operational analysis. They note however that [Qendresa] "is widely recognised as a Medicane".

Figure 4 Legends in all axes seem to mention the same pressure layers.

Yes, this was a typo and has been corrected.

Line 360: Could you please comment more on the reasons why cooling may be higher or lower depending on the case and/or location?

We added the following sentences: "Such a discrepancy with a storm of comparable intensity cannot be explained easily, and this is beyond the scope of the present work. A possible explanation could be the storm track affecting the same place by making a loop in the Gulf of Lion, resulting in a larger cooling. The difference can also come from a different oceanic preconditioning (their case occurred in May), with stronger stratification or a shallower mixed layer in the Gulf of Lion that amplifies cooling due to mixing/entrainment process."

References:

Gheusi, F., and Stein, J.: Lagrangian trajectory and air-mass tracking analyses with Meso-NH by means of Eulerian passive tracers, Techn. Doc., http://mesonh.aero.obs-mip.fr/mesonh/dir_doc/lag_m46_22avril2005/lagrangian46.pdf, accessed on 18/02/2019, 2005.

Schär, C., and Wernli, H.: Structure and evolution of an isolated semi-geostrophic cyclone. Quarterly Journal of the Royal Meteorological Society, 119(509), 57-90, 1993.

---

## Referee Report (RR1)

Review of **"Surface processes in the 7 November 2014 medicane from air-sea coupled high-resolution numerical modelling"** by Marie-Noëlle Bouin and Cindy Lebeaupin Brossier.

This paper aims at assessing the main role of the physical processes on the ocean surface and their interactions with the atmosphere for the Qendresa medicane event, which took place on 7 November southern Sicily. In order to understand the impact of the air-sea interactions, the authors have used a high-resolution (1.33 km) coupled model. Using this model, they also have assessed the role of different parameters affecting the air-sea heat exchanges. However, for this case study, main results shows that the features of this medicane (e.g., track, intensity or lifecycle) are not significant modified by the fact of using a couple-model.

I found this study very interesting and I think that these results will be useful for the community. The authors provide a detailed and extensive explanation of the physical processes (interaction between ocean surface and atmosphere) involved in the genesis, amplification and finally the decay phase of this medicane. Taking into consideration these results and how they can be useful for the community, I would like to see this paper published. However, I have some major concerns that should be addressed before it can be accepted for publication in the "Atmospheric Chemistry and Physics" Journal.

**Major Comment:**

I strongly recommend that the Authors resubmit this paper by including some or all of the following suggestions:

1) Drastic English Improvement. I think it is very important to improve significantly the English of this manuscript (sentences directly translated from the language of the authors to English, sentences too long and confusing, swap sentences, etc.). The fact the manuscript is not well-written distracts the reader from the content of the paper and also makes sometimes very difficult to understand what the authors are trying to communicate. The reader should not try to figure out what the authors are trying to explain. I understand that the Authors could not be native English speaker (as myself), so maybe a little help would be beneficial.

2) The second critical aspect I am concern about is the fact that the control simulation the authors are using does not verify accurately the observations (e.g., Fig.3 and Fig. 4), from the tracking and intensity point of view of the medicane. Is this simulation the best simulation they can produce with this couple model? How many simulations the authors have performed in order to obtain the 'best' simulation that resemble the observations? Sometimes, this process can carry out more than 30 simulations changing different parameters of the model setup... Following this, if our control simulation cannot describe

properly the observations, what is the point of using this simulation to describe the physical processes involved, in this case, in the medicane? In this case, the results and conclusions obtained do not properly describe the phenome we observe, they describe something else…

3) Also, I found this manuscript too long, taking into account that this case study has been studied and examined by other authors before and most of the information described in this paper confirm previous results, conclusions or explanations. I think that in some sections, the information provided is not relevant, so they could resume significantly some of these sections.

4) Finally, in the conclusion section, the authors include new discussion about different categories where medicanes could be sorted. I think that this information should be introduced in one of the first sections of the manuscript and not at the conclusions. Again, in this last section, authors should try to overview the content and not repeat excessively content explained before.

**Minor comments:**

The following are some suggestions that could help to improve the quality of the manuscript:

Introduction Section:

1) Page 1 (L1): I suggest to add some more references on Medicanes. This section is focussed on the description of the characteristics of Medicanes, but I feel that relevant references related with the definition of MEDICANES (acronym from Mediterranean Hurricane) are missing. Also, in the text the word medicane is related with mediterranean cyclones, and although the idea is clear, is not the definition used in the literature.

2) Page 1 (L27): "their radius ranges typically" -> "their radius typically ranges"

3) Page 1 (L28): "due to the enclosed **character** of the Mediterranean". What do you mean by character? I realized that the way of explaining the differences between Medicanes and TC (L25-34) is not very clear because of the use of long sentences separated by semicolons. I suggest to rephrase these ideas in a clearer way to facilitate the reader its comprehension.

4) Page 1 (L35-37) state that several studies documented different characteristics from medicanes, but only 1 reference is listed for each of these characteristics. I suggest to add more references.

5) Page 2 (L41): "impact ot the coastal reliefs in triggering deep convection…" -> Add references

6) Page 2 (L44-45): "It is nevertheless inadequate to …" -> This sentence it seems completely disconnected from the last sentence, which talks about the adapted version of the Hart Diagram. What do the authors mean stating that is inadequate to describe roles of upper-level and low-level processes? Do they mean that upper-level dynamics do not play a key role in the genesis of medicanes? I suggest to add some additional clarification of the meaning of this sentence.

7) Page 2 (L59-61): Add more references.

8) Page 2 (L65): "turning off selected processes in sensitivity experiments"-> In fact, the factor separation technique is a method of performing sensitivity experiments turning on/off different factors considered.

9) Page 3 (L84): "latent heat release **fed** at low level" -> It is not clear for me that this term can be used in this context.

Case study and simulations Section:

1) Page 4 (L125-126): "with high horizontal (1-2 km)" -> "with high **grid** horizontal and … resolutions"

2) Page 4 (L126): "present" -> "current"?

3) Page 4 (L127): "platforms" -> "centers"?

4) Page 5 (L162): "radiative transfers" -> "radiative transfer models"

5) Page 5 (L172): "ECMWF operational analyses" -> Please, provide more information about these fields.

6) Page 5 (L177): "with resolution 1.33 km" -> "with **grid** resolution 1.33 km"

7) Page 6 (L199): "configurations described previously" -> "configurations previously described"

Medicane lifecycle and coupling impact Section:

1) Page 7 (L266): "until its landfall" -> "until it makes landfall"?

2) Page 7 (L271): "collocated" -> "located"?, "placed"?

Role of surface fluxes and mechanisms Section:

1) Page 9 (L325-328): Too long sentence.

Figures:

Most of the figures are poor quality. I suggest to create .pdf or .eps format figures to increase quality of the manuscript.

Fig. 7: missing x and y labels.

Fig. 8-9-10. Misleading x-label. It should be replace with something such as: Time (hours UTC)

Fig. 11: Dashed line representing cross section region and the grey star should be highlighted. In addition, in the capture, the last line should be corrected to "Grey stars indicate the position…" instead of "**The** grey star**s** indicate the position…"

Fig. 12,14: missing x-label

Fig. 13, 16, 18: Coast lines are too width and difficult the visualization of the fields depicted. Please, improve this feature. Also, enlarge grey stars.

Fig. 17: Enlarge grey star.

---

## Author Response (AR2)

MINOR POINTS:

5   L104: compare -> compares
Change done.

L200: although not necessary, I suggest a Table to summarize the experiment setups.
Yes, we thought of it and finally decided against it because there are already 3 tables in the paper. But
10  we reformulated the description of the experiments to make it clearer.

L249: the choice of the radius of maximum wind as the radius to be used for the Hart diagram is not
convincing. In my opinion, this represents a strong underestimation of the cyclone extension. I suggest
to remove this point, or give a better motivation.
15  Considerations about the cyclone size, its time evolution and the period when it is representative of the
symmetry of the event have been added.

L269: weakens -> decreases
Change done

L278: … no change of the maximum wind compared to the previous period
This is not what we meant, and we realized that the sentence was misleading. It has been reformulated:
"maximum wind speed 8 m s-1 lower". Thank you for this.

25  L283, 396: … in Figure …
Done.

L325: ☐☐is missing
Done, thank you

L335: what do you mean with "neutral" wind?
The neutral wind (or equivalent neutral wind) is commonly used when computing surface turbulent
fluxes and corresponds to the wind speed obtained in neutral conditions in the surface layer (i.e. using a

logarithmic wind profile rather than stratification functions). We replaced "neutral wind" by "equivalent neutral wind" and added a reference.

Caption Fig. 10: please indicate that the figure refers to the NOCPL run
Done.

L373, 478: resulting from …
Done.

L374: precipitation areas …
Change done.

L377: add "(see Fig. 5a)" and the end of the paragraph
Change done.

L385: northeastwards …
Change done.

L408: H is controlled …
Change done

Caption Figure 11a: which level is shown?
First level of the model, this is now specified

Caption Figure 13: (d) refers to H, thus all panels should be changed
Done, thank you for checking this.

L435: "levels closest to the 1500 m" appear contrasting with what is reported in Figure 17 caption (the size of the symbols is inversely proportional to altitude between 0 and 1000m)
Yes, we realized that the caption was probably misleading. The final point of the particles is actually 1500 m, the symbol size in the figure in constant between 1000 and 1500 m asl, and increases when the height decreases. This is now specified in the caption.

L443: does the decrease in mixing ratio imply condensation and latent heating? In which way?

What we meant here is that the net increase of $\theta$ implies condensation. The decrease of the mixing ratio can be du to condensation or to other processes (precipitation).

70

L491: "and there is no real PV tower around the cyclone centre": be careful because this is not a general result, and it does not refer necessarily to cyclone of the first category. I think this sentence can make some confusion and I strongly suggest to remove it.

Done, thank you.

75

L492: "extending up to 800 hPa": really, the warm core extends also in the upper troposphere; do you mean 400 hPa?

Yes, we changed this.

80  L502: … is not due exclusively to … but also to …

Change done.

85

**Major comments**

1)  Drastic English Improvement. I think it is very important to improve significantly the English of this manuscript (sentences directly translated from the language of the authors to English, sentences too long and confusing, swap sentences, etc.). The fact the manuscript is not well-written distracts the reader from the content of the paper and also makes sometimes very difficult to understand what the authors are trying to communicate. The reader should not try to figure out what the authors are trying to explain. I understand that the Authors could not be native English speaker (as myself), so maybe a little help would be beneficial.

The text has been checked and rewritten almost entirely. We tried to improve the English, shorten the sentences and make it clearer overall. We hope that the present version is clear enough and does not prevent understanding the details of the scientific content.

2)  The second critical aspect I am concern about is the fact that the control simulation the authors are using does not verify accurately the observations (e.g., Fig.3 and Fig. 4), from the tracking and intensity point of view of the medicane. Is this simulation the best simulation they can produce with this couple model? How many simulations the authors have performed in order to obtain the 'best' simulation that resemble the observations? Sometimes, this process can carry out more than 30 simulations changing different parameters of the model setup... Following this, if our control simulation cannot describe properly the observations, what is the point of using this simulation to describe the physical processes involved, in this case, in the medicane? In this case, the results and conclusions obtained do not properly describe the phenome we observe, they describe something else…

We made 8 tests in order to improve our track, and using the results of Cioni et al (2018) for instance. Several tests concern the time of initialization of the run on the larger domain, and the best result (used in this work) was obtained starting the run at 18:00 UTC on 6 November. The large scale is known to largely control the track of tropical cyclone, so most of our tests were about the influence of the initial conditions. We also tested a configuration very close to the one of Cioni et al. (2018), with no grid nesting (using directly the ECMWF forcing on the smaller domain), and a horizontal resolution of 2 km. This simulation gave a much better track, but the intensification and tropical transition of the cyclone was missed. We think that this difference is due to a different physics in the models used. We also increased the number of vertical levels, as this is known to impact the track of tropical cyclones, but the difference was not significant. Besides, Di Muzio et al. (2019) showed that the predictability of this medicane 48 h to 24 h before its maximum intensity is low compared to other events (especially its track, and its central pressure). This confirms the results of our tests, i.e. obtaining both accurate track and intensity is challenging. We chose the configuration with the best intensity, and checked that the time evolution and impacts of the event is close to observation. Note also that an error of 48 to 54 nautical miles (89 to 100 km) represents a rather good score for the NHC tropical cyclone forecast at 36 h lead time (see https://www.nhc.noaa.gov/verification/verify4.shtml?). We are then confident in the capacity of our simulation to represent the physical processes and the effect of coupling during this event.

3) Also, I found this manuscript too long, taking into account that this case study has been studied and examined by other authors before and most of the information described in this paper confirm previous results, conclusions or explanations. I think that in some sections, the information provided is not relevant, so they could resume significantly some of these sections.

Some sections have been notably shortened, especially the sections presenting results (4 Role of surface fluxes and mechanisms) and the conclusion/discussion.

4) Finally, in the conclusion section, the authors include new discussion about different categories where medicanes could be sorted. I think that this information should be introduced in one of the first sections of the manuscript and not at the conclusions. Again, in this last section, authors should try to overview the content and not repeat excessively content explained before.

Part of the information presented in the discussion has been move to the introduction, following this recommendation.

**Minor comments**

Introduction

1) Page 1 (L1): I suggest to add some more references on Medicanes. This section is focussed on the description of the characteristics of Medicanes, but I feel that relevant references related with the definition of MEDICANES (acronym from Mediterranean Hurricane) are missing. Also, in the text the word medicane is related with mediterranean cyclones, and although the idea is clear, is not the definition used in the literature.

Yes, this has been clarified, and references have been added.

2) Page 1 (L27): "their radius ranges typically" -> "their radius typically ranges"

Change done

3) Page 1 (L28): "due to the enclosed **character** of the Mediterranean". What do you mean by character? I realized that the way of explaining the differences between Medicanes and TC (L25-34) is not very clear because of the use of long sentences separated by semicolons. I suggest to rephrase these ideas in a clearer way to facilitate the reader its comprehension.

What was meant is that as the Mediterranean is composed of several basins of small size, cyclone tracks rapidly encounter the coast and decay. We reformulated. We also rephrased the comparison between medicanes and TCs.

4) Page 1 (L35-37) state that several studies documented different characteristics from medicanes, but only 1 reference is listed for each of these characteristics. I suggest to add more references.

Done

5) Page 2 (L41): "impact ot the coastal reliefs in triggering deep convection..." -> Add references

Done

6) Page 2 (L44-45): "It is nevertheless inadequate to ..." -> This sentence it seems completely disconnected from the last sentence, which talks about the adapted version of the Hart Diagram. What do the authors mean stating that is inadequate to describe roles of upper-level and low-level processes? Do they mean that upper-level dynamics do not play a key role in the genesis of medicanes? I suggest to add some additional clarification of the meaning of this sentence.

This part has been removed from the Introduction.

7) Page 2 (L59-61): Add more references.

Done

8) Page 2 (L65): "turning off selected processes in sensitivity experiments"-> In fact, the factor separation technique is a method of performing sensitivity experiments turning on/off different factors considered.

Yes, thank you, this has been corrected.

9) Page 3 (L84): "latent heat release **fed** at low level" -> It is not clear for me that this term can be used in this context.

This has been reformulated

Case study and simulations

1) Page 4 (L125-126): "with high horizontal (1-2 km)" -> "with high **grid** horizontal and ... resolutions"

Change done

2) Page 4 (L126): "present" -> "current"?

Change done

3) Page 4 (L127): "platforms" -> "centers"?

Change done

4) Page 5 (L162): "radiative transfers" -> "radiative transfer models"

Change done

5) Page 5 (L172): "ECMWF operational analyses" -> Please, provide more information about these fields.

Done

6) Page 5 (L177): "with resolution 1.33 km" -> "with **grid** resolution 1.33 km"

Change done

7) Page 6 (L199): "configurations described previously" -> "configurations previously described"

Change done

Medicane lifecycle and coupling impact

1) Page 7 (L266): "until its landfall" -> "until it makes landfall"?

Change done

2) Page 7 (L271): "collocated" -> "located"?, "placed"?

What we meant here is that the upper level PV anomaly and the SLP minimum are aligned. We reformulated.

Role of surface fluxes

1) Page 9 (L325-328): Too long sentence.

We reformulated

Figures:

Most of the figures are poor quality. I suggest to create .pdf or .eps format figures to increase quality of the manuscript.

All the figures were redone in pdf format.

Fig. 7: missing x and y labels.

Corrected

Fig. 8-9-10. Misleading x-label. It should be replace with something such as: Time (hours UTC)

Change done

Fig. 11: Dashed line representing cross section region and the grey star should be highlighted. In addition, in the capture, the last line should be corrected to "Grey stars indicate the position..." instead of "**The** grey star**s** indicate the position..."

Changes done

Fig. 12,14: missing x-label

Change done

Fig. 13, 16, 18: Coast lines are too width and difficult the visualization of the fields depicted. Please, improve this feature. Also, enlarge grey stars.

Changes done

Fig. 17: Enlarge grey star.

Change done

[revised manuscript text omitted]